# Oral Health-Related Quality of Life in Children and Adolescents with a Traumatic Injury of Permanent Teeth and the Impact on Their Families: A Systematic Review

**DOI:** 10.3390/ijerph19053087

**Published:** 2022-03-06

**Authors:** Priyankaa Das, Lora Mishra, Debkant Jena, Shashirekha Govind, Saurav Panda, Barbara Lapinska

**Affiliations:** 1Department of Conservative Dentistry and Endodontics, Institute of Dental Sciences, Siksha ‘O’ Anusandhan, Bhubaneswar 751003, Odisha, India; priyankaa.das053@gmail.com (P.D.); debkantjena@soa.ac.in (D.J.); shashirekhag@soa.ac.in (S.G.); 2Department of Periodontics, Institute of Dental Sciences, Siksha ‘O’ Anusandhan, Bhubaneswar 751003, Odisha, India; sauravpanda@soa.ac.in; 3Department of General Dentistry, Medical University of Lodz, 92-213 Lodz, Poland

**Keywords:** oral health, quality of life, dental trauma, traumatic dental injury, permanent teeth, children, adolescent, well-being, systematic review, meta-analysis

## Abstract

The aim of this systematic review was to evaluate the impact of a traumatic dental injury (TDI) of permanent teeth in children and adolescents on their oral health-related quality of life (OHRQoL) as well as on their families. A bibliographic search in the biomedical databases (PubMed, Cochrane Library, MEDLINE) was limited to studies published between January 2000 and February 2021. The study selection criteria were cross-sectional, case control, or prospective clinical studies, which analyzed TDI before and after the treatment of permanent teeth in healthy children and adolescent, assessed their OHRQoL, and were written in English. The search found 25 eligible articles that were included in the study. The quality assessment of the studies was performed using the quality assessment checklist for survey studies in psychology (Q-SSP). The results indicated that a TDI of permanent teeth strongly influences the OHRQoL of children and adolescents, and the timely-performed dental management of a TDI allows for preventing further biological and socio-psychological impacts. Sociodemographic status, economic status, parent’s education, gender, age group, and type of schooling were determinants of the TDI impact on OHRQoL.

## 1. Introduction

Quality of life is described as “an individual’s perception of their place in life concerning goals, aspirations, standards, and concerns in the sense of the culture and values in which they reside” [1]. The definition of oral health-related quality of life (OHRQoL) refers to how oral health or disease affects an individual’s everyday work, well-being, and, as a result, their overall quality of life [2]. The quality of life is highly affected by their state of health. Physical and psychological constraints in the field of dentistry can directly affect eating, speech, social interaction, and self-esteem [3]. A traumatic dental injury (TDI) is an irreversible disease that is attracting more consideration from health practitioners at the moment [4]. A TDI, especially in children, is considered a severe health issue. Maxillary anterior teeth are the most affected teeth that cause physical, aesthetic, and psychological problems for children and their parents [5,6]. The quality of life is a complex process, and each person’s self-perceptions are shaped by their experiences, future expectations, dreams, and lifestyle [7]. Besides that, people change their view of their OHRQoL over time [8]. The second-most prevalent TDI is a crown fracture involving enamel and dentin (CFED). It is associated with trouble feeding, avoidance of smiling, sensitivity and discomfort, and a higher prevalence of adverse effects on OHRQoL [9,10]. Traumatic dental injuries to permanent teeth are more frequent than in primary dentition [11,12]. Dental injuries primarily concern the maxillary anterior teeth. Falls, sporting events, road traffic accidents, and bicycling are the most common causes of these injuries. Dental trauma predisposing factors may be related to the anatomical characteristics of the individual, such as increased overjet, insufficient lip coverage of the upper anterior teeth, etc. [13,14]. Home and school are areas where dental accidents frequently occur. It was observed that the place of injury was gender-related, i.e., the school followed by the home was the most common place of injury for boys, whereas this finding is vice versa for girls [15,16,17].

In everyday dental practice, treating dental injuries is not an ordinary condition. The result of the procedure is closely linked to the dentist’s expertise and skills and the medical assistance at the injury site. Thus, the dentist, parents, teachers, and coaches must have basic knowledge of dental trauma emergency management. However, the rareness of a TDI and the uncertainty of treatment prognosis, an individual with a traumatized tooth becomes a concern for the dentist. It is not a routine operation for most dentists and requires accurate diagnosis, appropriate emergency management, and correct follow-up treatment.

In the case of dental trauma, all treatment methods are aimed to mitigate undesired complications that may contribute to the loss of the tooth and the loss of the alveolar bone and thereby hinder the realization of a potential treatment plan. It is important to remember that traumatic dental injury care is vital for young people. It is essential to realize that treatment of a traumatic dental injury in a young patient is often complicated, unpredictable, expensive, and can continue for the remainder of his/her life. Since most of traumatic injuries in permanent dentition are between the ages of 10–12 years, dental trauma may have a lifelong effect on the child’s quality of life [18]. Therefore, the objective of this systematic review aims to assess the impact of a traumatic dental injury of permanent teeth on oral health-related quality of life and to assess the study quality using the Q-SSP checklist.

## 2. Materials and Methods

The review protocol was registered at PROSPERO (international prospective register of systematic reviews), bearing registration number CRD42021230281.

This review followed the Preferred Reporting Items for Systematic Reviews and Meta-Analyses (PRISMA) statement guidelines [19].

### 2.1. Search Strategy

The following structured question was outlined based on PICO (Patient or problem in question; Intervention of interest; Comparison of intervention; Outcomes): “Does trauma and treatment of traumatic injuries influence the OHRQoL of children and adolescents with a TDI and also how it impacts their family members?”.

The electronic search strategy is described in Table 1. A comprehensive electronic search for relevant articles was performed in the following databases: PubMed, Cochrane Library, MEDLINE, and Google Scholar. For all these databases, Boolean operators (OR, AND) were used to combine and narrow down searches that included appropriate MeSH terms, keywords, and other terms following the syntax rules of each database. All references selected in the search were saved in Mendeley Desktop software to remove the duplicates.

### 2.2. Study Selection

The literature search was limited to articles available in English and to those published between January 2000 and February 2021. Each article was assessed carefully and in detail.

Two independent reviewers (PD and LM) read abstracts and titles, and studies not pertaining to the research question were excluded. The remaining relevant studies’ full texts were read and analyzed independently. In this selection, if there was a disagreement of opinions, a third reviewer (DJ) was called to achieve a consensus.

The selection of studies was performed with no restrictions of place or year of publication. However, the restriction of language was applied, and only those articles written in English language were included. Titles and abstracts were analyzed to determine whether they fulfilled the inclusion criteria: (i) population: healthy children, adolescents, and family members; (ii) exposition: subjects experienced a TDI; (iii) outcome: impact on OHRQoL. The inclusion and exclusion criteria are depicted in Table 2.

Two reviewers (LM and SG) conducted the data extraction and collected the information independently. The relevant data of the included studies were extracted in detail, using Excel spreadsheet (Microsoft, Redmond, WA, USA, Version 2007). The extracted data included: title, journal name, year of publication, type of study, author, country, age group, instrument/application form, TDI index, sample size, tooth number, an association between TDI and OHRQoL, result, conclusion, publication, sample, country where the research was conducted, sample age, comparison, instrument applied, instrument purpose, TDI index, and type of treatment. Mean scores for the OHRQoL instruments (total scale and sub-scales) before and after treatment, *p*-value, and outcome were also identified.

### 2.3. Study Quality Assessment

The quality of the individual studies was assessed by one reviewer (PD) and independently checked for agreement by a second reviewer (LM). In case of disagreement, a third review author (DJ) was consulted. The quality assessment of the included studies was conducted using the quality assessment checklist for survey studies in psychology (Q-SSP) (Figure 1) [20], published in the year 2020, which includes 20 checklist items. The Q-SSP checklist has been developed to standardize responses to uniform quality assessment across researchers [20]. The quality was judged for each domain and is expressed as a percentage by dividing YES (Y) scores by the total (T) number of APPLICABLE items and multiplying by 100. When (T) = 20, then a Y/T ≥ 75% score may be considered acceptable quality. When (T) = 19, then a Y/T ≥ 73% score may be considered acceptable quality. When (T) = 18, then a Y/T ≥ 72% score may be considered acceptable quality. When (T) = 17, then a Y/T ≥ 70% score may be considered acceptable quality. If the report fails to attain a Y score for five items, it may be classified as having questionable quality. The assessment was added to an Excel spreadsheet and then imported into ROBVIS (Risk of Bias Visualization web app software).

## 3. Results

### 3.1. Selection of Studies

Figure 2 presents a flowchart of the systematic review process. The search in the selected databases allowed for the identification of 2677 articles. After removing duplicates, 2350 searches remained. Of these, 2297 were excluded after reading the titles and abstracts. From 53 remaining articles, 25 articles were finally selected after reading the full texts. Table A1 in Appendix A presents a list of the studies excluded after reading the full texts and the justification.

### 3.2. Characteristics of Studies

Most of the studies were cross-sectional [9,15,20,21,22,23,24,25,26,27,28,29,30], six studies were case control [31,32,33,34,35,36], and three studies were prospective clinical studies [37,38,39].

Nine studies [15,21,24,25,29,31,32,33,35] evaluated patients with TDI and no TDI, whereas eight studies [9,26,37,38,39,40,41,42] evaluated patients with TDI and after TDI treatment. One study [43] compared TDI and TDI with treatment needs. Another study [27] compared no TDI with TDI or without treatment needs. One study [22] evaluated TDI with no oral condition and TDI associated with dental caries, one study [23] evaluated mild/no TDI with severe TDI, one compared TDI with no treatment and TDI followed by treatment, two studies [28,30] evaluated patients with trauma, and one study [34] evaluated patients with TDI with unmet treatment needs and without TDI.

The included studies used different instruments to assess the OHRQoL (Table 3). From 25 included studies, the most widely used instrument for assessing OHRQoL of permanent teeth was the Child Perceptions Questionnaire (CPQ) (n = 15). The form of application most used was self-administered (n = 22). The Oral Impact on Daily Performances (OIDP) was used in five articles. The Family Impact Scale (FIS) was used in three articles, the Parental–Caregivers Perceptions Questionnaire (P-CPQ) in three articles, and the Oral Health Impact Profile (OHIP) in one study. One study used the National Research in Oral health (SBBrasil2010). Andreasen proposed the index for the TDI registry used by 15 articles. Seven articles used O’ Brien, one article used the Dental Trauma Index (DTI), and two articles used WHO 1997 for TDI registry. Regarding TDI association and impact on OHRQoL, 24 articles indicated an association, whereas one article indicated no association.

### 3.3. Analysis of Quality of the Studies

Risk of bias in included studies is presented in Figure 3 and Figure 4. Out of the 25 included articles, 21 articles are of acceptable quality scoring ≥75%, whereas 4 articles are of questionable quality with a score of <75%.

### 3.4. Synthesis of Results

Due to a high heterogeneity of the data, it was not possible to perform a meta-analysis for all the parameters used in the included studies; therefore, a qualitative assessment was performed. Table 4 shows the value of each domain and its impact on OHRQoL. 

## 4. Discussion

This review included 25 studies (Table 1) that assessed the impact of a traumatic dental injury of permanent teeth on the oral health-related quality of life (OHRQoL) in children and adolescent patients. The subjective evaluation of OHRQoL “reflects people’s comfort when eating, sleeping, and engaging in social interaction; their self-esteem; and their satisfaction concerning their oral health” [44]. With a growing emphasis on health promotion and illness prevention in health policy, OHRQoL has evolved to include positive and negative assessments of oral health and health outcomes [45]. As a result, oral health assessments might reveal both negative and positive effects on self-esteem and well-being.

We assessed the quality of the studies using the Q-SSP tool [20]. This tool helps researchers to perform a uniform quality assessment of survey studies in psychology across the globe. Using tools such as the Q-SSP checklist to evaluate study quality will raise the profile of reporting standards and drive greater precision in reporting the survey study methods. Researchers can use the tool to assess the quality of studies as an inclusion criterion in systematic reviews and meta-analyses. In addition, the tool may be used by professional clinicians, physicians, and practitioners wishing to evaluate the quality of psychological evidence that may inform their practice. It may also be helpful for educators to illustrate issues relating to study quality in research method courses.

Most of the studies used for evaluation of OHRQoL of patients with a TDI or a TDI with treatment needs were of acceptable quality, except for four studies [22,25,31,41], which had questionable quality (Figure 2). Therefore, the conclusion of these articles will be taken with caution. Seven tools were used in this systematic review. These were the P-CPQ tool, P-CPQ + CPQ(8–10), (11–14) + FIS, CPQ(8–10), CPQ(11–14)-16 short form and ten short forms, the Brazilian version of FIS, Oral Health Impact Profile (OHIP-14), Child-OIDP, and OIDP.

The most common tool used for the assessment of OHRQoL was CPQ(11–14) in 10 studies [15,25,26,27,29,31,33,41,42,43]. All questionnaire variations evaluated the impact of oral and orofacial conditions in children at symptomatic, functional, emotional, and social levels. To date, the CPQ has been translated, validated, and adapted to suit several languages and socio-cultural contexts, demonstrating its applicability and perfect psychometric properties on numerous clinical and epidemiological occasions [46].

The Child-OIDP was the second-most used tool for assessing the OHRQoL in children in five studies [9,28,34,35,36]. Child-OIDP and CPQ (11–14) differ in their aim and theoretical framework. The Child-OIDP has an advantage over the CPQ and other OHRQoL measures, as it specifies the different clinical causes of each oral impact [47,48]. The Child-OIDP has a greater sensitivity than CPQ in identifying the impact on the quality of life of schoolchildren with a TDI.

Due to much heterogeneity in the data, it was not possible to perform a meta-analysis for all the parameters used in studies; therefore, qualitative assessment was conducted. However, meta-analysis was possible for only two studies using the Child OIDP tool.

Most of the studies [24,25,29,31,32,33,35] that evaluated patients with TDI and no TDI revealed that patients with TDI have a negative impact on the OHRQoL, whereas in other studies [37,38,41] where TDI patients were compared based on whether they received treatment or not, they revealed that patients who received treatment had a positive impact on OHRQoL.

All the tools that analyzed the OHRQoL assessed the patient and their parents in various parameters, including sociodemographic status, economic status, parent’s education, gender, age group, and type of schooling. All these factors affected the OHRQoL of children, except for the type of schooling.

It was expected that a higher prevalence of TDI is in males compared to females [26], due to males being more engaged in sports and recreational activities involving physical contact. However, due to a change in social roles, adolescent females also pose an equal risk of TDI, as there is an increase in their participation in sports. Females are currently exposed to the same etiological factors. However, variations between genders may occur due to environmental, cultural, and behavioral factors, which are determinants of a stronger or weaker association between TDI and gender [49,50,51]. However, although four studies [25,26,28,31] out of eight [23,24,25,26,27,28,31,41] showed no association between gender and its impact on OHRQoL, another three studies [23,24,41] did imply that there is a strong association of TDI impact on OHRQoL among females compared to males. This outcome can be due to more significant aesthetic concerns of females than males, which negatively impacts their appearance [52,53,54].

All the studies expressed that the child perception of TDI impact on the OHRQoL does not change with age. Children between the age group of 8–10 years have criteria similar to those of children between 11–14 years regarding the self-perception of body image. To evaluate their appearance, children compare themselves to others of their age, and the judgment of peers exerts an influence on the development of self-esteem [55].

However, parents of the older children perceived a more significant reduction in their QoL than the parents of the 8–10-year-old group with TDI. This more remarkable impact on parents of the older children may be due to their children’s growing independence suddenly being reversed by the need for parental intervention and supervision [40].

Seven out of 10 studies evaluated another parameter that negatively impacts TDI associated OHRQoL: the family’s socioeconomic status of [9,21,24,27,29,30,34]. Due to their existing living conditions, which are usually less privileged and peripheries of urban areas, where facilities and quality healthcare are questionable, children are often exposed to unsafe environments. These underdeveloped areas increase accidents due to poorly-designed urban projects and neglected public spaces [56,57,58].

Even if parents wished to have the child’s condition treated, they cannot afford dental care at both private and public centers [59].

Parents’ education also impacts the OHRQoL of children with a TDI. Six studies [9,21,24,29,30,34] out of nine reported that fewer years of parents’ education level showed a negative impact on OHRQoL of children. The majority of parents in these studies had low education status. This reflects their lack of information, perception, and treatment needs associated with TDI and negatively influenced the child’s health behaviors.

Children with severe TDI also impacted the family regarding parental/family activity, parental emotions, family conflict, and financial burden [29,38,40]. Severe types of trauma more often affected the daily life of parents/caregivers. Parents/caregivers of adolescents with fractures involving the dentine or dentine/pulp reported more negative impact on parental/family activities than those with less severe TDI, such as enamel fracture [29]. A TDI is an unexpected event. More severe cases nearly always require urgent care and multiple searches for dental treatment, resulting in parents missing work and spending extra time taking care of their children. From these studies, it can be concluded that severe trauma not only affects the child in question, but it also affects the family.

Individual domains such as oral symptoms (OS), emotional well-being (EWB), social well-being (SWB), and functional limitations (FL) were mainly analyzed by different studies. These domains are individuals perception of TDI and their overall impact on OHRQoL. It was pretty evident that the most affected domains were EWB, OS, and FL. TDI was significantly affected by these three domains. The “emotional well-being” domain contains questions related to emotions such as sad, embarrassed, worried, upset, frustrated, angry, and concerned about what others think. Physical appearance and attractiveness play an essential role in social interactions and psychological well-being among adolescents between the ages of 11 and 14 [60]. Peer relationships are an important factor in an individual’s quality of life at this age [61]. Because the mouth is such a significant predictor of face attractiveness, any changes in dental features can have a detrimental or good impact on the quality of life [62]. The ‘oral symptoms’ domain contains questions about pain, wounds, mouth sores, bad breath, and food remains trapped in the mouth [38]. As this domain contains questionnaires related to lips, teeth, and jaws, the scores were high in this domain and thus affected the OHRQoL. The “functional limitations” domain contains questions related to difficulty with eating, biting, speaking, and sleeping. The overall cumulative effect of the individual domain significantly affected the TDI-associated OHRQoL.

Children with fractured teeth experienced more impacts on their daily living than children with no traumatic dental injury. Their actual daily basic performances such as ‘eating and enjoying food’, ‘cleaning teeth’, ‘smiling, laughing, and showing teeth without embarrassment’, ‘maintaining usual emotional state without being irritable’, and ‘enjoying contact with people’ significantly affected the OHRQoL when compared to children with no dental trauma experience [35].

It was observed that, after receiving the treatment of TDI, children were able to enjoy foods, smile, show one’s teeth without embarrassment, and socialize. Thus, dental treatment following a TDI is an important prevention strategy regarding biological and socio-psychological impacts [9]. Treatment of TDI improved the OHRQoL considerably.

## 5. Conclusions

Traumatic injuries to permanent dentition affect both a child and their caregivers or parents. These injuries affect both genders; however, adolescent girls tend to have a more negative impact on their OHRQoL than boys. A TDI and its severity significantly affect children and their families social and emotional well-being. Parents’ education and socioeconomic status play a significant role in providing care and treatment of TDIs in children. Treatment of TDIs improve the aesthetic and functional aspects of dentition and enhance the OHRQoL. Since the majority of studies used well-validated questionnaire tools and were of high quality, it can be concluded that the TDI impact on the OHRQoL is significant.

## Figures and Tables

**Figure 1 ijerph-19-03087-f001:**
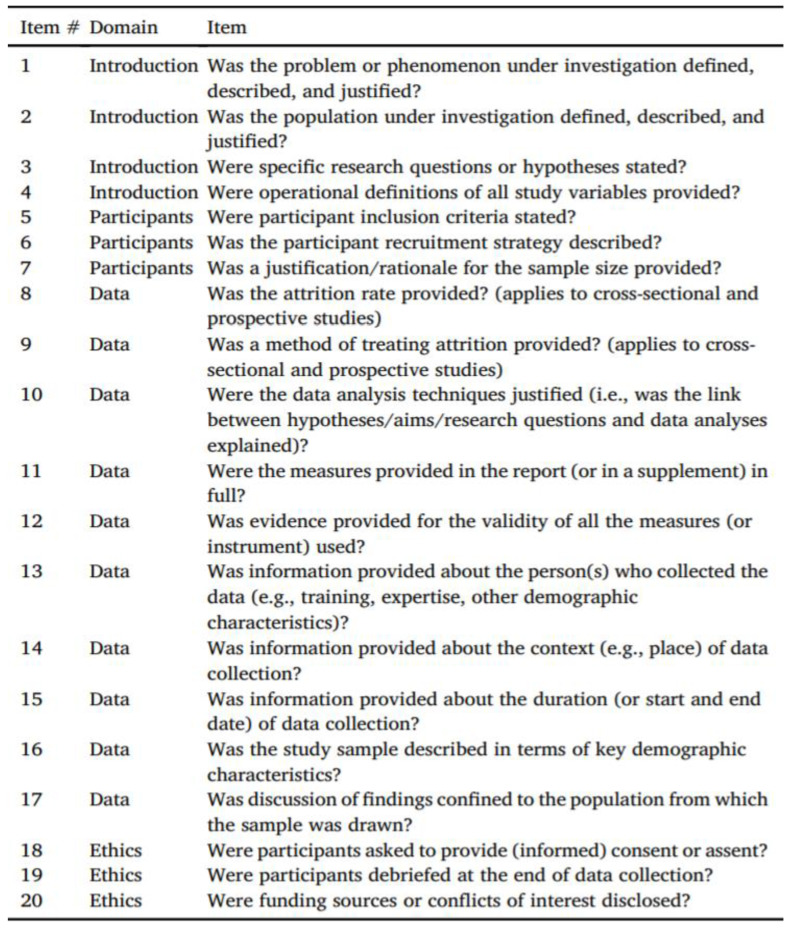
Q-SSP checklist for assessing quality of included studies.

**Figure 2 ijerph-19-03087-f002:**
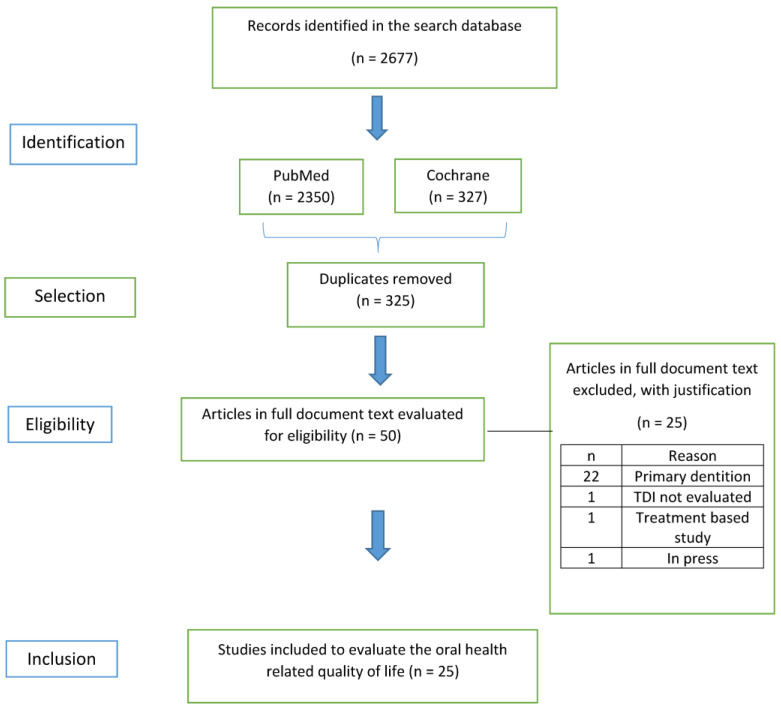
PRISMA 2020 flow diagram for systematic review that includes searches of databases.

**Figure 3 ijerph-19-03087-f003:**
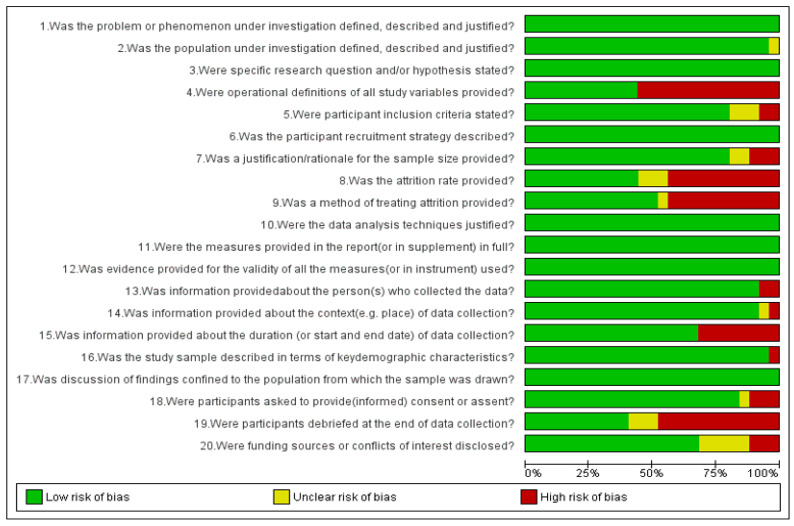
Quality assessment of studies using a QSSP tool graph: review authors’ judgements about each risk of bias item presented as percentages across all included studies.

**Figure 4 ijerph-19-03087-f004:**
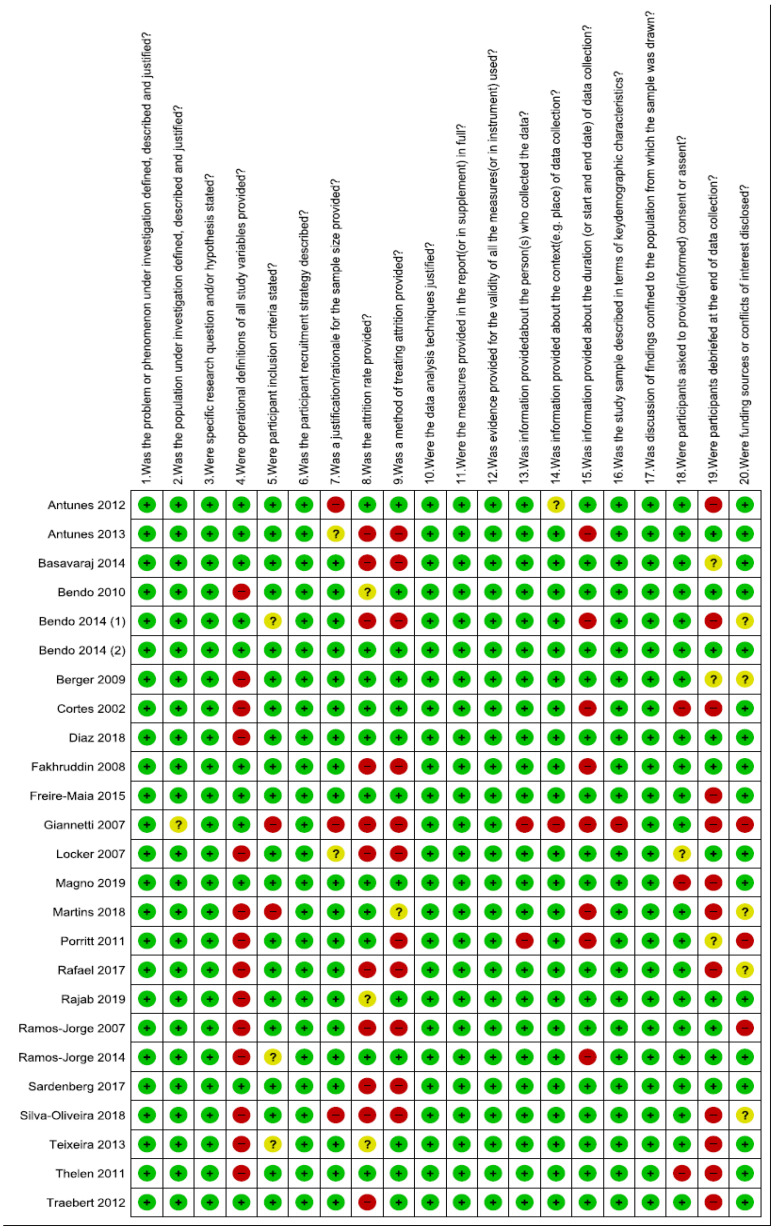
Quality assessment of included studies summary: review authors’ judgements about each risk of bias item for each included study.

**Table 1 ijerph-19-03087-t001:** Search strategy.

Search Strategy
#1 (Quality of life[MeSH Terms] OR Quality of life[Title/Abstract] OR QoL[Title/Abstract] OR OHRQoL[Title/Abstract] OR Early Childhood Oral Health Impact Scale[Title/Abstract] OR ECOHIS[Title/Abstract] OR Child Perceptions Questionnaire[Title/Abstract] OR CPQ 8–10[Title/Abstract] OR CPQ 11–14[Title/Abstract] OR Child-OIDP[Title/Abstract] OR SOHO[Title/Abstract] OR COHIP[Title/Abstract] OR PCPQ[Title/Abstract] OR Scale of Oral Health Outcomes[Title/Abstract] OR Psychology[Title/Abstract] OR Self esteem[Title/Abstract]
#2 (tooth injuries[MeSH Terms] OR tooth injuries[Title/Abstract] OR dental injuries[Title/Abstract] OR dental trauma[Title/Abstract] OR dentoalveolar trauma[Title/Abstract] OR tooth avulsion[Title/Abstract] OR Tooth Dislocation[Title/Abstract] OR Tooth Luxation[Title/Abstract] OR tooth intrusion[Title/Abstract] OR dental intrusion[Title/Abstract] OR tooth extrusion[Title/Abstract] OR tooth subluxation[Title/Abstract] OR Tooth Fractures[Title/Abstract] OR permanent teeth
Final search done:#1 and #2

**Table 2 ijerph-19-03087-t002:** Inclusion and exclusion criteria of selecting studies for systematic review.

Inclusion Criteria	Exclusion Criteria
Studies that analyzed TDI in healthy children and adolescents.	Studies on patients with medical conditions such as systemic diseases, syndromes, and craniofacial anomalies.
Studies that analyzed TDI before and after treatment of permanent teeth.	Studies on trauma to deciduous dentition, or where TDI was excluded and other oral health issues were addressed.
Studies must have assessed OHRQoL.	Studies that evaluated psychometric properties of instruments of OHRQoL or studies where only a single question of the questionnaire was used, evaluating only one domain.
Cross-sectional, case control, or prospective clinical study.	Case reports, review articles, systematic review articles, and book chapters.
Studies with abstract and full text in English language only.	

**Table 3 ijerph-19-03087-t003:** Data extraction of included studies.

Author/Year	Population Investigated	Age Group	Instrument	TDI Index	Association of TDI and OHRQoL	Conclusions	Funding
Diaz et al. (2018) [21]	Colombia	6–14 years	P-CPQ of COHRQoL	Andreasen	No	Children who studied at public schools were more likely to experience a negative impact on the emotional wellbeing and social wellbeing domains. There was no association between traumatic dental injuries and the perception of the impact of OHRQoL, but this may be due to the low prevalence of TDI in the sample.	None.
Antunes et al. (2012) [37]	Brazil	8–14 years	P-CPQ Brazilian version	Andreasen	Yes	The tooth most affected was the right maxillary central incisor (41.2%). The type of tissue most injured was dental tissue (54.8%). The most prevalent type of TDI was fracture of enamel and dentin (48.9%). It could be observed that the highest levels of impact and its reduction after treatment were in the group of trauma affecting both dental and support tissue. However, noticeable change over time could be identified (positive reduction) for all types of TDI, which denotes 100% of the population benefitting from trauma treatment.	DAB/SAS/MS (Department of Primary Care/Secretary of Health Care/Ministry of Health), DECIT/SCTIE/MS (Department of Science and Technology/Secretary of Science, Technology and Strategic Resources/Ministry of Health)—CNPq (The National Council for Scientific and Technological Development) and FAPERJ.
Magno et al. (2019) [38]	Brazil	8–14 years	CPQ8–10, CPQ11–14, P-CPQ, FIS	Andreasen	Yes	In general, children (aged 8–10 years) and adolescents (aged 10–14 years) presented with a reduction of the negative impact of OHRQoL following restorative treatment of CFED; however, the completion of the same treatment did not affect the OHRQoL of their families.	Coordenacao de Aperfeicoamnto de Pessoal de.
Berger et al. (2009) [40]	Canada	8–20 years	CPQ8–10, CPQ11–14, PPQ and FIS		Yes	Children and adolescents who sustain a dental injury severe enough to warrant splinting of the maxillary anterior teeth suffer an immediate decrease in their QoL. Results indicated that at one year, children are affected mostly in the emotional or social well-being domains, yet their parents exclusively reported one-year effects that were based on oral symptoms and functional limitations. Results from the emotional well-being component of the COHQoL questionnaire indicate that dental trauma continues to cause emotional distress and financial difficulties for the injured child and their parent one year later.	Dentistry Clinical Research Fund: Rhani Ghar Grotto Endowment.
Martins et al. (2018) [22]	Brazil	8–10 years	CPQ8–10	Andreasen	Yes	Children who presented with dental caries associated with TDI, as well as dental caries associated with malocclusion, were more likely to experience a high negative impact on their OHRQoL than those without any oral condition. Children with the three oral conditions were 2.01-fold more likely to experience a high negative impact on their OHRQoL (total score) than those without any oral health problems.	Not stated.
Sardenberg et al. (2017) [24]	Brazil	8–10 years	CPQ8–10	Andreasen	Yes	The mean CPQ8–10 score was 1.38-fold (95% CI: 1.17–1.63; *p* < 0.001) higher among the girls than boys, and children from families with a higher income had lower CPQ8–10 scores (RR: 0.67, 95% CI: 0.51–0.88; *p* < 0.004) than those from families with a lower income. Children who sought dental care due to pain or factors other than prevention, those with orofacial dysfunction, and those with a history of TDI also experienced a greater impact on OHRQoL.	Conselho Nacional de Desenvolvimento Científico e Tecnológico and Coordenação de Aperfeiçoamento de Pessoal de Nível Superior.
Freire-Maia et al. (2015) [23]	Brazil	8–10 years	CPQ8–10 Brazilian version	Andreasen	Yes	Girls had a 1.46-fold greater chance of presenting a high negative impact on OHRQoL and younger children had more chance of a high negative impact. Children with severe dental trauma (55.9%) reported more negative impact on OHRQoL than children with dental caries (44.4%) and/or accentuated anterior maxillary overjet (41.1%).	National Council for Scientific and Technological Development (CNPq), the Ministry of Science and Technology, and the State of Minas Gerais Research Foundation (FAPEMIG), Brazilian Coordination of Higher Education (CAPES), Brazil.
Silva-Oliveira et al. (2018) [25]	Brazil	12 years	CPQ11–14—ISF:16 short form	Andreasen	Yes	The central incisors were the most affected teeth. TDI was associated with an overjet equal to or greater than 3 mm. There was also an association of the negative impact on oral health-related quality of life, among patients who presented with TDI, in the social well-being and emotional well-being subscales. No association between TDI and socioeconomic factors was observed.	Not stated.
Rajab et al. (2019) [26]	Jordan	12 years	CPQ11–14 Arabic version	Andreasen	Yes	When each of the 16 items of the CPQ11–14 was considered, higher impacts were reported by children who had untreated TDI. The mean scores of the 16 items of the CPQ11–14 were higher in the group of untreated TDI than those in both the group treated TDI and the group with absence of trauma. The results of the present study confirm the negative impact of untreated TDI on QoL of schoolchildren.	Not stated.
Bendo et al. (2010) [15]	Brazil	11–14 years	CPQ11–14 Brazilian version	Andreasen	Yes	Children with untreated TDI were 1.2-fold (95% CI = 0.9–1.6) more likely to feel “upset” and 1.2-fold (95% CI = 0.9–1.7) more likely to have “avoided smiling/ laughing” than children without TDI. In the comparison of children with treated fractures and those without TDI, there was no association to the overall CPQ11–14—ISF: 16 score (Fisher = 0.610). Dental pain and difficulty chewing were more prevalent among children with treated teeth than those with no TDI, but this difference did not achieve statistical significance (*p* > 0.05).	National Council for Scientific and Technological Development (CNPq), Ministry of Science and Technology, and the State of Minas Gerais Research Foundation (FAPEMIG), Brazil.
Bendo et al. (2014) (1) [31]	Brazil	11–14 years	CPQ11–14 Brazilian version	Andreasen	Yes	Age was not associated with impact on adolescents’ OHRQoL. However, adolescents diagnosed with fractures involving dentin and/or pulp, untreated dental caries, and malocclusion had a greater chance of presenting high negative impact on OHRQoL. The results demonstrated that adolescents diagnosed with fractures involving dentin and/or pulp had a 2.40-fold greater chance of presenting high negative impact on QHRQoL than those without evidence of TDI.	Coordination for the Improvement of Higher-Level Education Personnel (CAPES), the National Council for Scientific and Technological Development (CNPq), and the State of Minas Gerais Research Foundation (FAPEMIG), Brazil.
Porrit et al. (2011) [41]	UK	7–17 years	CPQ11–14 – ISF:16 short form	Andreasen	Yes	The results revealed that girls were more likely to report a higher level of impact on their OHRQoL and HRQoL than boys following traumatic injury to their permanent incisors.	Not stated.
Traebert et al. (2012) [43]	Brazil	11–14 years	CPQ11–14 – ISF:16 short form	O’Brien	Yes	Enamel fractures were the most common form of TDI, while adhesive restoration was the most common form of treatment needed for TDI. This study showed a statistically significant and independent association between TDI and OHRQoL among Brazilian 11–14-year-old schoolchildren.	Grant from FAPESC – Fundacao de Apoio a.
Dame-Texeira et al. (2013) [27]	Brazil	12 years	CPQ11–14- ISF:16 short form	O’Brien	No	Individuals presenting TDI with treatment needs experienced a higher average CPQ11–14 score than individuals with no TDI or with TDI without treatment needs. The main finding was that schoolchildren affected by TDI and needing clinical intervention had significantly higher adjusted mean CPQ11–14 scores for function impairment than those with no TDI or affected by TDI with no treatment needs, indicating a significant but limited effect on quality of life. Where no overall association was observed between TDI and OHRQoL, a domain-specific analysis revealed significant association between TDI and function impairment. Schoolchildren presenting with TDI with clinical treatment needs (e.g., restorations, crowns, root canal therapy) had a 1.2-fold higher adjusted mean CPQ11–14 score than the reference group (no TDI/no treatment needs).	None.
Antunes et al. (2013) [32]	Brazil	10–15 y	CPQ11–14 – ISF:16 short form	WHO 1997	Yes	Children and adolescents with traumatic dental injury were more likely to have a greater impact on their life than those with no injuries. Traumatic dental injury actually affects the quality-of-life of children and adolescents and, consequently, it is not enough to treat only its signs and physical symptoms. In fact, oral symptoms but also functional limitations and emotional and social well-being should be considered.	Not stated.
Locker et al. (2007) [42]	Canada	11/12 years 13/14 years	CPQ11–14—10 short form	Dental Trauma Index	Yes	Over one third, 37.5%, showed evidence of injury to the anterior dentition (DTI codes of 1–5), with 15.3% having one or more teeth with severe injury (DTI codes of 2–5). Children from low-income households had higher scores on a short form of the CPQ11–14 than children from high-income households, indicating poorer oral health-related quality of life.	Grant from the Ontario Ministry of Health.
Fakhruddin et al. (2008) [33]	Canada	12–14 years	CPQ11–14—10 short form	Dental Trauma Index	Yes	Children with untreated dental injuries were approximately three times more likely to report difficulty chewing than those without injury. Subjects with untreated dental trauma were approximately three times more likely to avoid smiling or laughing and four times more likely to report not wanting to talk to other children compared with uninjured controls. The impact of dental trauma to upper incisors on social well-being was greater than on functional and psychological well-being in this sample of 12–14-year-old schoolchildren. Those with untreated dental injuries experienced a higher risk of negative social impact on their daily living than those without injury.	Grant from the Ontario Ministry of Health.
Bendo et al. (2014) (2) [29]	Brazil	11–14 years	FIS Brazilian version	Andreasen	Yes	TDI severity was directly associated with an impact on the family’s QoL, especially regarding parental/family activities. Parents/caregivers of adolescents with fractures involving the dentine or dentine/pulp reported more negative impact on parental/family activities than those with less severe TDI, such as enamel fracture.	Coordination for the Improvement of Higher-Level Education Personnel (CAPES), Ministry of Education, and the State of Minas Gerais Research Foundation (FAPEMIG), Brazil.
Gianenetti et al. (2007) [39]	Italy	Under 18 years	OHIP-14	Andreasen	Yes	It was a single tooth avulsed in 63.3% of the population, 49.5% was central incisor. Adverse impacts on OHRQoL were reported much more frequently among patients who got into failure of replantation compared with patients who got into successful replantation. The findings show that if patients got into tooth avulsion, then their quality of life is adversely affected.	Not stated.
Bomfim et al. (2017) [30]	Brazil	12 years	National Research in Oral Health (SBBrasil2010)		Yes	Regarding occlusal characteristics, crowding in at least one segment was associated with trauma in the maxillary teeth and in mandibular teeth. Crowding in two segments increased the chances of fracture. The spacing/diastema between the arches was a risk factor for enamel fractures, fractures in mandibular teeth, and for any fracture analyzed. The presence of a diastema and mandibular overjet was not associated with any type of TDI. Maxillary overjet (greater than 3 mm) was associated with all fractures in maxillary teeth. Anterior open bite was a protective factor for enamel fractures in maxillary teeth and any analyzed TDI.	Not stated.
Ramos-Jorge et al. (2014) [9]	Brazil	11–14 years	Child-OIDP	O’Brien	Yes	Schoolchildren with untreated TDI experienced a greater negative impact on quality of life in comparison with those without TDI. This impact was significant regarding eating and smiling. No significant differences were found on the Child-OIDP between schoolchildren with treated TDI and those without TDI. The association between untreated TDI and impact on quality of life in the present study was stronger for ‘eating and enjoying food’ and ‘smiling and showing teeth’.	Brazilian fostering agencies the Coordination of Higher Education (CAPES), Ministry of Education, and the State of Minas Gerais Research Foundation (FAPEMIG).
Thelen et al. (2011) [34]	Albania	16–19 years	OIDP	O’Brien	Yes	The overall impact prevalence of OIDP among cases was significantly higher (88.4%) than for the controls (58.9%). The most prevalent impact was ‘smiling and showing teeth without embarrassment’ which was reported by cases 78.9% and their controls 31.6%. The second-most prevalent impact was ‘enjoying contact with people’. TDI with unmet treatment needs in this sample of adolescents are associated with reduced OHRQoL. Compared to adolescents with no history of TDI, those affected by TDI with unmet treatment needs are at greater risk of suffering impacts on OHRQoL in the form of OIDP.	Department of Clinical Dentistry and the Centre for International Health, University of Bergen.
Basavaraj et al. (2014) [28]	India	12 and 15 years	Child-OIDP	WHO	Yes	Impacts on eating were the most prevalent (45.3%). The prevalence of impacts on cleaning teeth (42.3%) and smiling (40.1%) were also relatively high. There is a strong association between clinical dental indicators and oral impacts in children.	None.
Cortes et al. (2002) [35]	Brazil	12–14 years	OIDP	O’Brien	Yes	The prevalence of oral impacts, measured by the OIDP index, was higher for children with untreated fractured teeth than for children with non-fractured teeth. For both groups of children, the most prevalent OIDP impact was ‘smiling, laughing, and showing teeth without embarrassment’, with the proportion being higher for cases (55.9%) than for controls (13.2%).	Grant from Conselho Nacional de Pesquisa (CNPq).
Ramos-Jorge et al. (2007) [36]	Brazil	12–14 years	OIDP	O’Brien	Yes	The impact prevalence was greater in the case group for nearly all the appraised activities. In the previous six months, 40% of the teenagers with a history of treatment for enamel dentin fractures had at least one negatively affected daily activity, and 16.9% of the teenagers without a history of trauma were found to have some oral impact on their daily lives. The most affected activities in decreasing order were showing teeth when smiling, eating, speaking, maintaining a stable emotional state, and cleaning the mouth. Sleeping, doing school tasks, practicing sports, and going out with friends were all mentioned as activities that no adolescent reported as having an impact. Adolescents with aesthetically-treated enamel dentin fractures were more likely than those who had never experienced dental trauma to present oral impact on daily activities. Treatment for coronary fractures does not completely eradicate the impact of trauma on the adolescents daily lives, but it does help to mitigate it.	Not stated.

Legend: OHRQoL—oral health-related quality of life; P-CPQ—Parental–Caregiver Perception Questionnaire; CPQ—Child Perceptions Questionnaire; FIS—Family Impact Scale; CFED—crown fracture involving enamel and dentin; OHIP—Oral Health Impact Profile; COHQoL—Child Oral Health Quality of Life; ISF—Impact Short Form; DTI—Dental Trauma Index; Child-OIDP—Child-Oral Impacts on Daily Performances; OIDP—Oral Impact on Daily Performances.

**Table 4 ijerph-19-03087-t004:** Data extraction of included studies—the value of each domain and its impact on OHRQoL.

Author/Year	Value of Each Domain									Results		
Diaz et al. (2018) [21]	**P-CPQ**	**Mean**		**SD**				**Range**		Significant association (*p* < 0.05) between oral symptoms and mother’s education and family income; emotional wellbeing domain and dental caries experience; social wellbeing domain and children’s education, number of siblings; total PCPQ and members in family and dental caries. Children from public schools and children who had dental caries experience (RR = 1.28; *p* = 0.04 and RR = 1.37; *p* = 0.018, respectively) had a negative impact on total PCPQ scores. Public school-going children were more likely to experience negative impact on the emotional wellbeing and social wellbeing domains (*p* < 0.05). Children whose mothers had an educational level < 10 years and children who had dental caries experience showed positive and negative impact on the oral symptoms domain, respectively (RR = 0.75, *p* = 0.02 and RR = 1.22, *p* = 0.04, respectively).		
	**OS**	3.88		3.5				0–20				
	**FL**	3.43		4.17				0–24				
	**EWB**	2.09		3.90				0–30				
	**SWB**	3.09		6.15				0–48				
	**Total score**	12.49		14.04				0–90				
Antunes et al. (2012) [37]	**B-P-CPQ**	**A1**				**A2**				The group of trauma affecting both dental and support tissue had the highest levels of impact (A1) and the greatest reduction in impact following therapy (A2). Positive reduction after receiving the treatment over time was observed for all types of TDI, indicating that 100% of the population benefitted from trauma treatment. Post-treatment oral symptoms did positively affect the OHRQoL. However, out of all the domain scores, oral symptoms had the lowest impact on OHRQoL. The FL domain had the highest impact on OHRQoL. The functional limitations drastically improved the OHRQoL. The EWB and SWB positively improved the OHRQoL after treatment. Overall post-treatment of TDI significantly improved the OHRQoL. Out of all four domains tested, EWB and FL improved and resulted in the highest impact on OHRQoL.		
	***p*-value < 0.01**	**Mean (SD)**		**Median**		**Mean (SD)**		**Median**				
	**OS(0–24)**	3.36 (3.11)		3.00		0.05 (0.31)		0.00				
	**FL(0–32)**	9.83 (6.50)		9.50		1.38 (2.78)		0.00				
	**EWB(0–28)**	9.12 (6.60)		10.00		0.17 (0.70)		0.00				
	**SWB(0–40)**	7.74 (6.41)		7.00		1.07 (1.63)		0.00				
	**Total(0–124)**	30.05 (17.39)		27.50		2.67 (4.02)		2.00				
Magno et al. (2019) [38]	**CPQ(8–10)**	**BT**		**AT**				***p*-value**		OS domain: CPQ(8–10): There was significant improvement in oral symptoms after treatment. CPQ(11–14): This observation was not statistically significant. This implied that for children between 11–14 years, oral symptoms perception did not affect the OHRQoL. Overall assessment: This indicated that overall parents perception on OHRQoL in the oral symptom domain improved after the child received the treatment. FL domain: The FL domain score for all three questionnaires did not show significant difference in domain score (*p* > 0.05). This result signifies that functional limitations before and post treatment, both in children and parent perception, did not statistically or significantly impact the OHRQoL. EWB domain: For CPQ(8–10), there was no impact on emotional well-being. CPQ(11–14) scores implied that EWB improved post treatment and had positive impact on OHRQoL in children between the age group of 11–14 years. *p*-CPQ scores indicated no significant difference. SWB domain: The CPQ(8–10), CPQ(11–14), and P-CPQ scores indicated that there was no statistical significant difference in score in all three groups. This indicated that SWB did not statistically impact the OHRQoL. The cumulative scores of all domains indicated that there was significant improvement in OHRQoL for CPQ(8–10), CPQ(9–14), and P-CPQ after receiving the treatment of TDI. Familiar Impact Scale (FIS) scores: TDI among children involving enamel and dentine fracture did not impact family perspective.		
	**OS**	5.3 (3.4)		2.7 (2.9)				0.0003 b				
	**FL**	2.6 (3.5)		1.7 (2.0)				0.4498 b				
	**EWB**	1.1 (2.8)		0.3 (0.7)				0.4990 b				
	**SWB**	2.8 (3.5)		1.5 (1.1)				0.0843 b				
	**TOTAL**	10.8 (10.0)		6.5 (4.5)				0.0065 b				
	**CPQ(11–14)**											
	**OS**	3.2 (2.6)		2.4 (1.8)				0.37 a				
	**FL**	2.0 (1.6)		1.2 (1.6)				0.2049 b				
	**EWB**	1.1 (1.6)		0.0 (0.0)				0.0431 b				
	**SWB**	2.5 (3.5)		1.3 (1.7)				0.1083 b				
	**TOTAL**	8.8 (5.4)		5.4 (2.7)				0.0486 a				
	**P-CPQ**											
	**OS**	4.5 (3.2)		3.6 (2.9)				0.0455 b				
	**FL**	5.4 (4.4)		3.9 (3.5)				0.1213 b				
	**EWB**	5.4 (6.5)		3.5 (6.6)				0.0534 b				
	**SWB**	5.4 (4.9)		4.7 (6.5)				0.1482 b				
	**TOTAL**	20.7 (14.1)		15.7 (16.6)				0.0259 b				
	**FIS**											
	**PE**	5.1 (4.6)		5.7 (5.3)				0.8456				
	**FC**	1.7 (2.2)		1.3 (2.3)				0.2805				
	**FA**	1.2 (1.8)		1.0 (1.8)				0.1823				
	**TOTAL**	7.9 (7.4)		8.0 (7.9)				0.5850				
Berger et al. (2009) [40]		n		**CPQ(8–10)**		**PPQ(8–10)**		**FIS**		6 months: After receiving the treatment, the post6-month follow up of the COHRQoL score was improved in all domains. However, the parental COHRQoL scores were dependent upon the initial COHRQoL scores at *p* = 0.03 (ANCOVA), but the patient scores were not dependent upon the initial score at *p* = 0.12 (ANCOVA). The PPQ was statistically significant when compared with baseline values. 12 months: The parental 12-month results indicated that scores were dependent upon the initial scores (*p* = 0.001, ANCOVA). The child 12-month results (CPQ8–10, CPQ11–14) were also dependent upon the baseline COHRQoL scores (*p* = 0.005, ANCOVA). At 12 months, both age groups of children/adolescents reported lasting effects in each of the four CPQ domains, but their parents only saw lasting effects in two domains (oral symptoms and functional limitations), and they did not offer a single response in the emotional and social well-being domains for both age groups. One year after the injury, the parents of 11–14 year-old patients noticed a significant ongoing effect on their personal QoL. The high initial parental PPQ scores suggest that TDI has a significant impact on the parents’ QoL. FIS: There was no significant differences between the FIS scores for the 8–10 years and 11–14 years age group across all time periods. The initial high parental FIS scores suggest that TDI has a significant impact on the parents’ QoL. Parents of the older children perceived their child’s pain as being greater than the pain reported by the patient, and younger children perceived the initial injury as more painful than the older group.		
	**Initial**	11		31.2 (13.3)		34.8 (18.6)		13.1 (6.4)				
	**6 months**	10		20.6 (14.8)		20.6 (21.8)		9.7 (8.2)				
	**12 months**	8		17.5 (12.3)		15.9 (12.0)		7.6 (6.1)				
		n		**CPQ11–14**		**PPQ11–14**		**FIS**				
	**Initial**	12		29.3 (10.9)		38.8 (22.6)		9.8 (6.9)				
	**6 months**	11		19.8 (12.2)		28.0 (17.7)		7.6 (5.6)				
	**12 months**	9		16.7 (9.3)		27.4 (18.3)		7.2 (6.0)				
Martins et al. (2018) [22]	**Variables** **Mean (SD)** ***p*-value < 0.001**	**OS**		**FL**	**EWB**	**SWB**		**TOTAL**		OS domain: TDI showed no statistical significant difference with the group that had no oral condition. However, dental caries associated with TDI had a high negative impact on OHRQoL. FL domain: The dental caries children scores were statistically significant with children with TDI at *p* = 0.001. EWB domain: In this domain, TDI had no significant impact on OHRQoL. SWB: In this domain, TDI had no significant impact on OHRQoL. Children who presented with dental caries associated with TDI were more likely to experience a high negative impact on their OHRQoL, as shown by the total score, than those without any oral condition. The presence of dental caries and its association with TDI were significantly associated with all CPQ(8–10) subscales at *p* < 0.05.		
	**No conditions**	3.84 (3.07)		1.87 (2.72)	3.15 (4.08)	2.89 (4.63)		11.61 (11.88)				
	**Dental caries**	5.31(3.52)		3.02 (3.59)	4.82 (4.55)	4.36 (5.42)		17.50 (14.31)				
	**Malocclusion**	4.14 (3.40)		2.37(3.02)	4.83(4.83)	4.18 (6.03)		15.49 (14.54)				
	**TDI**	3.91(3.16)		1.53(2.13)	3.39 (4.14)	3.06 (4.20)		12.03 (11.52)				
Freire-Maia et al. (2015) [23]	**Domain**	**OR**		**95%**		** *p* **				Effect of Gender on OHRQL: Girls had a 1.46-fold greater chance of presenting with a negative impact on OHRQoL. Effect of trauma status on OHRQoL: Children with severe dental trauma reported a more negative impact on OHRQoL than children with dental caries and malocclusion involving increased anterior overjet. Effect of individual domain score of TDI on OHRQOL: Severe trauma was significantly associated with a negative impact on overall quality of life (55.9%). Trauma did not significantly affect the OS and FL domain, but SWB and EWB egatively affected the OHRQOL in the bivariate analyses.		
	**OS**	2.67		1.31–5.46		0.005						
	**SWB**	2.93		1.46–5.90		0.002						
	**EWB**	2.61		1.31–5.20		0.005						
Sardenberg et al. (2017) [24]	**CPQ8–10** **Subscales**	**Number of items**		**Mean (S.D.)**		**Possible range**		**Observed range**		Effect of gender on OHRQoL: OHRQoL was significantly associated (*p* < 0.05) with sex. The CPQ(8–10) score was 1.38 times (95% CI: 1.17–1.63; *p* < 0.001) higher among girls than boys. This signified that there was a more negative impact on OHRQoL among girls when compared to boys. Socioeconomic status and education level of parents: Negative impact on children’s OHRQoL was significantly associated with a lower parent’s/ guardian’s schooling and lower family income. Effect of TDI and orofacial dysfunction on OHRQoL: A significant negative impact on OHRQoL associated with a history of TDI (RR: 1.39; 95% CI: 1.15–1.69) and orofacial dysfunction (RR: 1.62; 95% CI: 1.30–2.02) was seen. Effect of access to dental care: The children who reported difficulty in having access to dental care and only sought dental care due to pain or factors other than prevention (RR: 1.41; 95% CI: 1.18–1.68) were more likely to experience a negative impact on OHRQoL.		
	**OS**	5		5.15 (3.64)		0–20		0–19				
	**FL**	5		2.84 (3.52)		0–20		0–18				
	**EWB**	5		3.51 (4.47)		0–20		0–20				
	**SWB**	10		2.45 (4.50)		0–40		0–28				
	**Overall**	25		13.95 (13.12)		0–100		0–76				
Silva-Oliveira et al. (2018) [25]	**Subscale**	**Mean**		**Independent variable**		**TDI present** **n (%)**		***p*-value**		Effect of gender on OHRQoL: There was no association between TDI and gender. Impact of TDI alone on OHRQoL: There were statistically significant values indicating that TDI was associated with a high impact on OHRQoL (OR = 1.61 (95% CI: 1.08–2.39)) in children. Effect of mother’s education on OHRQoL: Mother’s educational level was not related to TDI and did not affect the OHRQoL in children. Effect of socioeconomic status on TDI: Monthly household income was not related to TDI and did not affect the OHRQoL in children. Effect of type of school on OHRQoL: The type of school was not related to TDI and did not affect the OHRQoL in children. TDI was significantly associated with the results of social and emotional well-being subscales. Overall, TDI exerted a negative impact on the OHRQoL of the adolescents analyzed.		
	**OS**	2.65		Low impact High impact		65 (27.0) 108 (31.1)		0.277				
	**FL**	1.70		Low impact High impact		74 (26.8) 99 (31.7)		0.191				
	**EWB**	1.66		Low impact High impact		77 (24.6) 96 (34.9)		0.006				
	**SWB**	1.27		Low impact High impact		47 (20.2) 126 (35.5)		<0.001				
	**TOTAL**	12.54		Low impact High impact		69 (24.2) 107 (35.3)		0.003				
Rajab et al. (2019) [26]	**Domain**			**Mean (S.D.)**		**Range Observed**				Socioeconomic status: The results of simple logistic regression showed that social class had no significant impact on overall QoL. Effect of gender: Gender had no significant impact on overall QoL. There was a significant impact of untreated TDI on overall OHRQoL. The impact of oral health of the study sample on QoL was much greater on OS and EWB than FL and SWB. Children with untreated TDI in this study risked a higher negative impact on their daily living than those without TDI. Children with untreated TDI had significant poorer overall OHRQoL than those with treated injured teeth and those without trauma. The results of the present study confirmed the negative impact of untreated TDI on QoL of schoolchildren.		
	**OS**			4.45 (3.25)		0.00–16.00						
	**FL**			2.91 (3.13)		0.00–16.00						
	**EWB**			3.54 (3.73)		0.00–16.00						
	**SWB**			2.37 (3.09)		0.00–15.00						
	**TOTAL**			13.27 (11.41)		0.00–55.00						
Bendo et al. (2010) [15]	**Domain**	**Untreated TDI**		**Absence of TDI**		**Unadjusted PR (95% CI)**		***p*-value †- Chi square test** **‡-Fisher’s test**		There were no statistically significant differences between children with untreated TDI and those without TDI in terms of the overall CPQ(11–14) scores. There was no association between the overall CPQ11–14—ISF:16 score (Fisher = 0.610) in children with treated fractures and those without TDI.		
	**OS**											
	**Pain** **CPQ11–14 = 0** **CPQ11–14 ≥1**	81 (37.0) 138 (63.0)		541 (40.5) 796 (59.5)		1 1.1 (0.8–1.5)		0.330				
	**Mouth sores** **CPQ11–14 = 0** **CPQ11–14 ≥1**	85 (38.8) 134 (61.2)		475 (35.5) 862 (64.5)		1 0.8 (0.6–1.1)		0.348 †				
	**FL**											
	**Difficulty chewing** **CPQ11–14 = 0** **CPQ11–14 ≥1**	128 (58.4) 91 (41.6)		772 (57.7) 565 (42.3)		1 0.9 (0.7–1.3)		0.844 †				
	**Difficulty eating/drinking hot/cold foods** **CPQ11–14 = 0** **CPQ11–14 ≥1**	84 (38.4) 135 (61.6)		455 (34.0) 882 (66.0)		1 0.8 (0.6–1.1)		0.212 †				
	**EWB**											
	**Felt irritable/frustrated** **CPQ11–14 = 0** **CPQ11–14 ≥1**	138 (63.0) 81 (37.0)		827 (61.9) 510 (38.1)		1 0.9 (0.7–1.2)		0.743 †				
	**Upset** **CPQ11–14 = 0** **CPQ11–14 ≥1**	118 (53.9) 101 (46.1)		795 (59.5) 542 (40.5)		1 1.2 (0.9–1.6)		0.120 †				
	**Concerned with what others think** **CPQ11–14 = 0** **CPQ11–14 ≥1**	107 (48.9) 112 (51.1)		548 (41.0) 789 (59.0)		1 0.7 (0.5–0.9)		0.029 †				
	**SWB**											
	**Avoided smiling/laughing** **CPQ11–14 = 0** **CPQ11–14 ≥1**	141 (64.4) 78 (35.6)		939 (70.2) 398 (29.8)		1 1.2 (0.9–1.7)		0.082 †				
	**Teased/called names** **CPQ11–14 = 0** **CPQ11–14 ≥1**	151 (68.9) 68 (31.1)		913 (68.3) 424 (31.7)		1 0.9 (0.7–1.3)		0.845 †				
	**Other children asked questions** **CPQ11–14 = 0** **CPQ11–14 ≥1**	125 (57.1) 94 (42.9)		832 (62.2) 505 (37.8)		1 1.2 (0.9–1.6)		0.146 †				
	**Overall** **CPQ11–14 = 0** **CPQ11–14 ≥1**	5 (2.3) 214 (97.7)		19 (1.4) 1318 (98.6)		0.6 (0.2–1.6)		0.368 ‡				
	**Domain**			**Treated TDI**		**Unadjusted PR (95% CI)**		***p*-value**				
	**OS**											
	**Pain** **CPQ11–14 = 0** **CPQ11–14 ≥1**			20 (31.2) 44 (68.8)		1 1.4 (0.8–2.5)		0.142 †				
	**Mouth sores** **CPQ11–14 = 0** **CPQ11–14 ≥1**			23 (35.9) 41 (64.1)		1 0.9 (0.5–1.6)		0.947 †				
	**FL**											
	**Difficulty chewing** **CPQ11–14 = 0** **CPQ11–14 ≥1**			32 (50.0) 32 (50.0)		1 1.3 (0.8–2.2)		0.221 †				
	**Difficulty eating/drinking hot/cold foods** **CPQ11–14 = 0** **CPQ11–14 ≥1**			25 (39.1) 39 (60.9)		1 0.8 (0.4–1.3)		0.407 †				
	**EWB**											
	**Felt irritable/frustrated** **CPQ11–14 = 0** **CPQ11–14 ≥1**			43 (67.2) 21 (32.8)		1 0.7 (0.4–1.3)		0.390 †				
	**Upset** **CPQ11–14 = 0** **CPQ11–14 ≥1**			41 (64.1) 23 (35.9)		1 0.8 (0.4–1.3)		0.464 †				
	**Concerned with what others think** **CPQ11–14 = 0** **CPQ11–14 ≥1**			28 (43.8) 36 (56.2)		1 0.8 (0.5–1.4)		0.661 †				
	**SWB**											
	**Avoided smiling/laughing** **CPQ11–14 = 0** **CPQ11–14 ≥1**			45 (70.3) 19 (29.7)		1 0.9 (0.5–1.7)		0.989 †				
	**Teased/called names** **CPQ11–14 = 0** **CPQ11–14 ≥1**			48 (75.0) 16 (25.0)		1 0.7 (0.4–1.2)		0.258 †				
	**Other children asked questions** **CPQ11–14 = 0** **CPQ11–14 ≥1**			31 (48.4) 33 (51.6)		1 1.5 (1.1–2.8)		0.027 †				
	**Overall** **CPQ11–14 = 0** **CPQ11–14 ≥1**			1 (1.6) 63 (98.4)		1 0.9 (0.1–6.8)		0.610 ‡				
Bendo et al. (2014) (1) [31]	**Variables**	**Case**		**Control**		**Unadjusted OR (95% CI)**		***p*-value**		Effect of gender on OHRQoL: Gender did not affect the OHRQoL. Effect of age on OHRQoL: Age was not associated with impact on adolescents’ OHRQoL. Effect of type of school: Type of school had no impact on TDI. Effect of type of dental trauma on OHRQoL: There was strong association between more severe TDI (fractures involving dentin and/or pulp) and poorer OHRQoL among adolescents. However, mild TDI (enamel fractures only) and restored fractures were not associated with negative impact on OHRQoL. Overall effect of TDI using the CPQ tool on OHRQoL: The overall CPQ(11–14) showed that TDI appeared to affect an adolescent’s OHRQoL. There is a strong association between the severity of TDI and OHRQoL.		
	**Traumatic dental injuries**											
	**Without injuries**	340 (84.0)		694 (85.7)		1.00						
	**Restored fracture**	20 (4.9)		25 (3.1)		1.63 (0.89–2.98)		0.110				
	**Enamel fracture only**	23 (5.7)		73 (9.0)		0.64 (0.40–1.05)		0.075				
	**Fracture involving dentin/pulp**	22 (5.4)		18 (2.2)		2.50 (1.32–4.71)		0.005				
Porrit et al. (2011) [41]	**Domain**	**N**		**Baseline mean (SD)**		**Follow-up mean (SD)**	**Wilcoxon test Z^±^**	**Sig change**		Effect of gender on OHRQoL: Gender was found to be a significant predictor of children’s OHRQoL. The results revealed that girls were more likely to report impacts on their OHRQoL (F (1) = 6.58, *p* < 0.05) than boys. Effect of age on OHRQoL: There were no significant interaction effects between gender and age or age when sustained injury or deprivation. 6-month follow-up: At the 6-month follow up, school functioning and functional limitations remained the areas of children’s OHRQoL that had the most impacts. A total of 62.9% of children reported improvement in their OHRQoL, whereas 30% of children had a negative impact, and 7.1% had no impact on OHRQoL at the 6-month follow up. Children with high levels of OHRQoL at baseline were more likely to report high levels of OHRQoL at follow up. Effect of individual domain on OHRQoL: With respect to OHRQoL, children were most likely to report difficulties related to OS, FL, and EWB domains. Social impacts remained the least affected area, within the child’s OHRQoL, at the 6-month follow up.		
	**Overall**	70										
	**OS**	70		4.2 (3.0)		3.0 (2.5)	Z = −3.13 **	↑				
	**FL**	70		4.3 (3.6)		3.2 (3.1)	Z = −3.18 **	↑				
	**Emotional impacts**	70		3.8 (3.9)		2.9 (3.2)	Z = −2.22 *	↑				
	**Social impacts**	70		3.2 (3.4)		2.9 (2.8)	Z = −1.03	↔				
Traebert et al. (2012) [43]	**Mean CPQ11–14 (SD)**									OS: Significant association was observed between TDI and OS domain at *p*-value 0.036. FL: Significant association was observed between TDI and FL domain at *p*-value 0.013. EWB: Significant association was observed between TDI and EWB domain at *p*-value 0.030. Overall score: There were significant associations between the prevalence of one or more adverse impacts occurring often/very often through the overall CPQ11–14 scale and TDI (*p* = 0.007). A prevalence ratio of 1.79 (95% CI 1.16–2.76) of one or more adverse impacts occurring often/very often in schoolchildren with TDI was found, compared to those without TDI. TDI has a strong association with appearing to affect school children’s OHRQoL.		
		**Overall**		**OS**		**FL**		**EW**		**SW**	**95% CI**	
	**Schoolchildren with TDI**	14.6 (8.6)		4.7 (2.3)		3.4 (3.0)		3.7 (3.1)		2.8 (2.7)	62.1 (50.4–73.8) **	
	**Schoolchildren without TDI**	9.6 (7.5)		3.8 (2.4)		2.3 (2.4)		1.8 (2.5)		1.7 (2.1)	44.0 (38.7–49.3)	
	***p*-value**	0.019 *		0.026 *		0.016 *		0.031 *		0.869 *	0.019 **	
	**All children**	12.4 (9.2)		4.1 (2.6)		2.8 (2.9)		3.4 (3.5)		2.1 (2.5)	46.5 (41.6–51.4)	
Dame-Texeira et al. (2013) [27]	**Domain**	**No TDI/No treatment**		**Treated**		**Treatment need**		**Total**		Effect of gender on OHRQoL: There was significant association. Effect of socioeconomic status on OHRQoL: There was significant association. Effect of individual domain on OHRQoL: OS: The OS domain was not associated with TDI. This indicated that TDI did not affect the OHRQoL. FL: The FL domain was significantly associated with TDI with treatment needs, whereas no association was observed with treated TDI. EWB: The EWB domain was not associated with TDI. This indicated that TDI did not affect the OHRQoL. SWB: The SWB domain was not associated with TDI. This indicated that TDI did not affect the OHRQoL. Overall score: Individuals presenting TDI with treatment needs experienced a higher average CPQ11–14 score than individuals with no TDI or with TDI without treatment needs (RR = 1.2; 95% CI = 1.0–1.4). The overall CPQ11–14 score was not associated with TDI. This indicated that TDI did not affect the OHRQoL.		
		Mean (95% CI)		Mean (95% CI)	*p*-value	Mean (95% CI)		*p*-value				
	**OS**	4.15 (3.74–4.56)		4.48 (3.58–5.37)	0.198	4.44 (3.65–5.23)		0.183	4.18 (3.77–4.59)			
	**FL**	2.96(2.64–3.29)		3.39 (2.03–4.76)	0.385	3.63 (3.11–4.16)		0.132	3.02 (2.70–3.35)			
	**EWB**	2.90 (2.48–3.31)		2.79 (1.89–3.69)	0.714	2.81 (1.94–3.69)		0.771	2.89 (2.49–3.28)			
	**SWB**	2.33 (1.98–2.69)		2.03 (1.43–2.64)	0.427	2.84 (2.11–3.56)		0.210	2.36 (2.03–2.68)			
	**Overall** **CPQ11–14**	12.35 (10.98–13.72)		12.70 (10.58–14.83)	0.652	13.74 (11.70–15.78)		0.476	12.46 (11.21–13.72)			
Antunes et al. (2013) [32]	**CPQ11–14 Domain**	**Case Group** **Mean ± SD**		**Control group** **Mean ± SD**				***p*-value**		Effect of individual domain on OHRQoL: OS: There was a statistically significant difference between the case and control group. This indicated that children with TDI had a negative impact on OHRQoL. FL: Children with TDI had a negative impact on OHRQoL. EWB: Children with TDI had a negative impact on OHRQoL. SWB: Children with TDI had a negative impact on OHRQoL. Overall score: Children with TDI had a negative impact on OHRQoL. Children and adolescents with a traumatic dental injury were more likely to have a greater impact on their life than those with no injuries.		
	**OS**	3.82 ± 2.60		1.30 ± 2.02				<0.01				
	**FL**	5.29 ± 4.03		1.33 ± 1.94				<0.01				
	**EWB**	5.00 ± 6.34		0.24 ± 1.22				<0.01				
	**SWB**	3.47 ± 4.36		0.21 ± 0.59				<0.01				
	**TOTAL**	17.59 ± 14.01		3.09 ± 4.42				<0.01				
Locker et al. (2007) [42]	**Clinical indicator**			**Mean CPQ11–14 score**		***p*-value**				Associations were significant for all variables except school grade and mother’s educational attainment. Both variables denoting the socioeconomic status of the household in which the child participants lived (annual household income, receipt of government income support) indicated that children from lower-income households had the highest CPQ11–14 short form scores. In the higher income group, there were no differences in CPQ11–14 scores for children with or without severe injury to the anterior dentition. However, the differences were significant for children in the lower-income group. Children from low-income households had higher scores on a short form of the CPQ11–14 than children from high-income households, indicating poorer oral health-related quality of life.		
	**Incisors with DTI codes 1–5**											
	**None**			12.7		NS						
	**One**			13.4								
	**Two or more**			13.7								
	**Incisors with DTI codes 2–5**											
	**None**			12.7		<0.001						
	**One**			13.6								
	**Two or more**			16.4								
Fakhruddin et al. (2008) [33]	**Dimensions and items**	**Case (n = 92)**		**Control (n = 92)**		**Unadjusted odds ratio ** (95% CI)**		**Adjusted odds ratio ** (95% CI)**				
		Untreated dental injury n (%)		No dental injury n (%)								
	**OS**											
	**Pain** **CPQ11–14 = 0** **CPQ11–14 = 1**	54 (58.7) 38 (41.3)		59 (64.1) 33 (35.9)		1.31 (0.68–2.52)		1.54 (0.71–3.36)				
	**FL**											
	**Sleep disturbances** **CPQ11–14 = 0** **CPQ11–14 = 1**	86 (93.5) 6 (6.5)		84 (91.3) 8 (8.7)		0.75 (0.26–2.16)		1.29 (0.39–4.16)				
	**Chewing difficulty** **CPQ11–14 = 0** **CPQ11–14 = 1**	59 (64.1) 33 (35.9)		70 (76.1) 22 (23.9)		2.00 (0.97–4.12)		2.86 (1.13–7.26) *				
	**EWB**											
	**Shy or embarrassed** **CPQ11–14 = 0** **CPQ11–14 = 1**	72 (78.3) 20 (21.7)		76 (82.6) 16 (17.4)		1.27 (0.64–2.49)		1.71 (0.78–3.75)				
	**Concerned with what others think** **CPQ11–14 = 0** **CPQ11–14 = 1**	63 (68.5) 29 (31.5)		76 (82.6) 16 (17.4)		2.00 (1.03–3.89) *		2.07 (0.96–4.47)				
	**SWB**											
	**Low concentration in school** **CPQ11–14 = 0** **CPQ11–14 = 1**	79 (85.9) 13 (14.1)		82 (90.1) 9 (9.9)		1.50 (0.61–3.67)		1.80 (0.67–4.87)				
	**Avoid smiling/laughing** **CPQ11–14 = 0** **CPQ11–14 = 1**	72 (78.3) 20 (21.7)		83 (90.2) 9 (9.8)		2.38 (1.04–5.43) *		3.09 (1.12–8.50) *				
	**Did not want to talk to other children** **CPQ11–14 = 0** **CPQ11–14 = 1**	79 (85.9) 13 (14.1)		88 (95.7) 4 (4.3)		3.25 (1.06–9.97) *		3.84 (1.12–13.18) *				
	**Did not want to spend time with other children** **CPQ11–14 = 0** **CPQ11–14 = 1**	85 (92.4) 7 (7.6)		90 (97.8) 1 2 (2.2)		3.50 (0.73–16.84)		5.12 (0.85–30.76)				
	**Teased by other children** **CPQ11–14 = 0** **CPQ11–14 = 1**	79 (85.9) 13 (14.1)		85 (92.4) 7 (7.6)		1.86 (0.74–4.65)		2.19 (0.78–6.18)				
	**Overall CPQ11–14** **CPQ11–14 = 0** **CPQ11–14 = 1**	33 (35.9) 59 (64.1)		44 (47.8) 48 (52.2)		1.58 (0.89–2.81)		1.80 (0.93–3.48)				
	**Dimensions and items**	**Case (n = 43)**		**Control (n = 43)**		**Unadjusted odds ratio ** (95% CI)**		**Adjusted odds ratio ** (95% CI)**				
		**Restored injury** **n (%)**		**No dental injury** **n (%)**								
	**OS**											
	**Pain** **CPQ11–14 = 0** **CPQ11–14 = 1**	24 (55.8) 19 (44.2)		24 (55.8) 19 (44.2)		1.00 (0.42–2.40)		1.17 (0.40–3.43)				
	**FL**											
	**Sleep disturbances** **CPQ11–14 = 0** **CPQ11–14 = 1**	41 (95.3) 2 (4.7)		37 (86.0) 6 (14.0)		0.33 (0.07–1.65)		0.16 (0.02–1.32)				
	**Chewing difficulty** **CPQ11–14 = 0** **CPQ11–14 = 1**	27 (62.8) 16 (37.2)		35 (81.4) 8 (18.6)		2.60 (0.93–7.29)		4.16 (1.08–16.12) *				
	**EWB**											
	**Shy or embarrassed** **CPQ11–14 = 0** **CPQ11–14 = 1**	31 (72.1) 12 (27.9)		35 (81.4) 8 (18.6)		2.33 (0.60–9.02)		2.14 (0.37–12.31)				
	**Concerned with what others think** **CPQ11–14 = 0** **CPQ11–14 = 1**	32 (74.4) 11 (25.6)		34 (79.1) 9 (20.9)		1.40 (0.44–4.41)		2.01 (0.39–10.29)				
	**SWB**											
	**Low concentration in school** **CPQ11–14 = 0** **CPQ11–14 = 1**	39 (90.7) 4 (9.3)		40 (95.2) 2 (4.8)		2.00 (0.37–10.91)		1.81 (0.23–14.15)				
	**Avoid smiling/laughing** **CPQ11–14 = 0** **CPQ11–14 = 1**	36 (83.7) 7 (16.3)		39 (90.7) 4 (9.3)		2.00 (0.50–7.99)		1.67 (0.26–10.82)				
	**Did not want to talk to other children** **CPQ11–14 = 0** **CPQ11–14 = 1**	38 (88.4) 5 (11.6)		41 (95.3) 2 (4.7)		2.50 (0.49–12.89)		1.16 (0.13–10.75)				
	**Did not want to spend time with other children** **CPQ11–14 = 0** **CPQ11–14 = 1**	38 (88.4) 5 (11.6)		41 (95.3) 2 (4.7)		2.50 (0.49–12.89)		0.74 (0.09–5.49) `				
	**Teased by other children** **CPQ11–14 = 0** **CPQ11–14 = 1**	38 (88.4) 5 (11.6)		39 (90.7) 4 (9.3)		1.33 (0.29–5.96)		2.54 (0.29–22.33)				
	**Overall CPQ11–14** **CPQ11–14 = 0** **CPQ11–14 = 1**	16 (37.2) 27 (62.8)		18 (41.9) 25 (58.1)		1.20 (0.52–2.78)		1.43 (0.52–3.88)				
Bendo et al. (2014) (2) [29]	**Variables**	**Overall B-FIS Robust RR (95% CI)**	**Parental/ family activity** **Robust RR (95% CI)**	**Parental emotions** **Robust RR (95% CI)**		**Family conflict Robust RR (95% CI)**		**Financial burden** **Robust RR (95% CI)**		Effect of individual domain: Parental/ family activity: Parents/caregivers of adolescents who had suffered a fracture involving the dentine or dentine/pulp had higher scores on the parental/family activity subscale than those whose adolescents were diagnosed with an absence of TDI or enamel fracture alone. Greater social vulnerability had a negative impact on families’ QoL regarding parental/family activity. The severity of the TDI was significantly associated with negative impacts on the parental activity. Parental emotions: Greater social vulnerability had a negative impact on families’ QoL regarding parental emotions. The severity of the TDI was significantly associated with negative impacts on the parental emotion subscale. Family conflict: Greater social vulnerability had a negative impact on families’ QoL regarding the family conflict subscales. The severity of the TDI was significantly associated with negative impacts on the family conflict subscales. Financial burden: There was absence of impact on the financial burden subscale, which reflects the fact that TDI is not considered a disease by most parents. Overall score: Parents/caregivers of adolescents who had suffered a fracture involving the dentine or dentine/pulp had higher scores on overall B-FIS than those whose adolescents were diagnosed with an absence of TDI or enamel fracture alone. Greater social vulnerability had a negative impact on families’ QoL regarding the overall B-FIS. Adolescents with a fracture of dentine or dentine/pulp had a 44%-higher chance of increasing their overall B-FIS score by one point (RR = 1.44; 95%zx CI; 1.10–1.88) than those without TDI. A fracture involving dentin or dentin/pulp was associated with a greater likelihood of a negative impact on family’s QoL.		
	**TDI** **absent**	1.00	1.00	1.00		1.00		1.00				
	**Enamel fracture alone**	0.96 (0.77–1.18)	1.04 (0.82–1.32)	0.87 (0.67–1.12)		0.98 (0.73–1.30)		0.78 (0.52–1.16)				
	**Fracture involving dentine or dentine/pulp**	1.44 (1.10–1.88) **	1.45 (1.09–1.94) *	1.45 (1.03–2.04) *		1.46 (1.01–2.11) *		1.26 (0.79–2.00)				
Gianenetti et al. (2007) [39]	**Age** **% under 18 years old**			72.27 (73)						Adverse impacts on OHRQoL were reported much more frequently among patients who got into failure of replantation compared with patients who got into successful replantation. If patients got into tooth avulsion, then their quality of life was adversely affected.		
	**Sex** **% Male** **% Female**			63.4 (64) 36.6 (37)								
	**Tooth Avulsed** **% central incisors ** **% lateral incisors ** **% more than one element**			49.5 (50) 13.8 (14) 36.7 (37)								
	**Time since last dental visit** **% visited in last month**			39.6 (40)								
Bomfim et al. (2017) [30]	**Trauma**	**n**		**%**		**CI 95%**						
	**Maxillary**	1344		18.56		17.68		19.47		Effect of family income on OHRQoL: Income level had no association with TDI. This indicated that family income did not impact the OHRQoL. Effect of parent’s education on OHRQoL: Parents education was not associated with TDI outcome. This indicated that parental education did not impact the OHRQoL. Effect of trauma on OHRQoL: Enamel fractures were risk factors for feelings of shame among children (OR 1.27 and 95%CI: 1:05–1:53) and were significantly associated with embarrassment of smiling, whereas dentine/enamel fractures were risk factors for dissatisfaction with their teeth or for feeling embarrassed of smiling and messing up with the study. This type of TDI was also associated with the unadjusted coefficient used to report difficulty with eating. This indicated that dentin fracture or fractures involving pulp impacted the OHRQoL negatively. Mandibular tooth fractures did not affect the quality of life of 12-year-old Brazilian children. The greater the severity of the TDI, the greater its impact on OHRQoL. TDI causes aesthetic, emotional, and functional problems in patients that might be irreversible in some cases.		
	**Mandibular**	391		5.4		4.9		5.9				
	**Enamel**	1378		19.03		18.1		20				
	**Dentine**	271		3.7		3.32		4.2				
	**Pulp exposition**	22		0.3		0.2		0.4				
	**Avulsion**	12		0.17		0.1		0.3				
Ramos-Jorge et al. (2014) [9]	**Variables**	**Without TDI** **n (%)**		**Untreated TDI** **n (%)**		**Treated TDI** **n (%)**		** *p* **		Effect of mother’s education on OHRQoL: There was a statistically significant difference for mother’s schooling in comparison of schoolchildren without TDI and those with treated TDI. Effect of individual items on OIDP: Children with untreated TDI experienced a greater negative impact on QoL in comparison with those without TDI in eating and enjoying food and smiling and showing teeth. No impact on OIDP was seen in all children in the treated TDI group for cleaning mouth, speaking, sleeping, and relaxing. No impact on OIDP was seen in all children in the no TDI group, untreated TDI, or in treated TDI group in maintaining usual emotional state and carrying out school-related tasks. Enjoying contact with people: Impact on OIDP was seen in 0.2% of children in the without TDI group and no impact on OIDP was seen in all children in the untreated TDI group and treated TDI group. Overall: Children with untreated TDI experienced a greater negative impact on QoL in comparison with those without TDI.		
	**Eating and enjoying food** **Child-OIDP = 0** **Child-OIDP ≥1**	419 (95.4) 20 (4.6)		148 (90.2) 16 (9.8)		59 (90.8) 6 (9.2)		Without vs. Untreated = 0.016 Without vs. Treated = 0.128				
	**Cleaning mouth** **Child-OIDP = 0** **Child-OIDP ≥1**	428 (97.5) 11 (2.5)		162 (98.8) 2 (1.2)		65 (100.0)		Without vs. Untreated = 0.530 Without vs. Treated = 0.3742				
	**Speaking** **Child-OIDP = 0** **Child-OIDP ≥1**	434 (98.9) 5 (1.1)		159 (97.0) 5 (3.0)		65 (100.0)		Without vs. Untreated = 0.145 Without vs. Treated = 1.000				
	**Sleeping and relaxing** **Child-OIDP = 0** **Child-OIDP ≥1**	435 (99.1) 4 (0.9)		163 (99.4) 1 (0.6)		65 (100.0) -		Without vs. Untreated = 1.000 Without vs. Treated = 1.000				
	**Smiling and showing teeth** **Child-OIDP = 0** **Child-OIDP ≥1**	395 (90.0) 44 (10.0)		130 (79.3) 34 (20.7)		56 (86.2) 9 (13.8)		Without vs. Untreated < 0.001 Without vs. Treated = 0.348				
	**Maintaining usual emotional state** **Child-OIDP = 0** **Child-OIDP ≥1**	439 (100.0) -		164 (100.0) -		65 (100.0) -		*				
	**Carrying out school-related tasks** **Child-OIDP = 0** **Child-OIDP ≥1**	439 (100.0) -		164 (100.0) -		65 (100.0) -		*				
	**Enjoying contact with people** **Child-OIDP = 0** **Child-OIDP ≥1**	438 (99.8) 1 (0.2)		164 (100.0) -		65 (100.0) -		Without vs. Untreated = 1.000 Without vs. Treated = 1.000				
	**Overall OIDP** **Child-OIDP = 0** **Child-OIDP ≥1**	364 (82.9) 75 (17.1)		115 (70.1) 49 (29.9)		52 (80.0) 13 (20.0)		Without vs. Untreated < 0.001 Without vs. Treated = 0.563				
Thelen et al. (2011) [34]	**Variables**	**Cases n (%)**		**Controls n (%)**		**OR-unadjusted** **(CI = 95%)**		**OR-adjusted** **(CI = 95%)**		Effect of individual items on OHRQoL: Eating and enjoying food: A total of 46.3% of cases and 34.2% of controls had an impact on OIDP. Cleaning mouth: A total of 28.4% of cases and 28.4% of controls had an impact on OIDP. Speaking: A total of 3.2% of cases and 8.9% of controls had an impact on OIDP. Sleeping and relaxing: A total of 18.9% of cases and 18.4% of controls had an impact on OIDP. Smiling and showing teeth: A total of 78.9% of cases and 31.6% of controls had an impact on OIDP. This item showed the most prevalent impact on OHRQoL. Statistical significant difference was observed between cases and controls. Maintaining usual emotional state: A total of 31.6% of cases and 20% of controls had an impact on OIDP. A significant difference was found between cases and controls at *p*-value <0.05. Carrying out school-related tasks: A total of 4.2% cases of and 11.6% of controls had an impact on OIDP. Enjoying contact with people: A total of 66.3% of cases and 23.2% of controls had an impact on OIDP. This item had the second-most prevalent impact, which was significantly more in cases than controls at *p*-value <0.001. Statistical significant difference was observed between cases and controls. Overall score: The overall impact prevalence of OIDP among cases was significantly higher (88.4%) than for the controls (58.9%) (*p* < 0.001). There was a significantly greater probability of perceiving an oral impact on daily life among cases than controls. TDI with unmet treatment need in this sample of adolescents was associated with reduced OHRQoL. Compared to adolescents with no history of TDI, those affected by TDI with unmet treatment need were at greater risk of suffering impacts on OHRQoL in the form of OIDP.		
	**Eating and enjoying food** **OIDP = 0** **OIDP ≥1**	51 (53.7) 44 (46.3)		125 (65.8) 65 (34.2)		1 1.7 (1.01–2.78) *p* < 0.05		1 1.01 (0.6–2.1)				
	**Cleaning your mouth** **OIDP = 0** **OIDP ≥1**	68 (71.6) 27 (28.4)		136 (71.6) 54 (28.4)		1 0.3 (0.08–1.2)		1 0.2 (0.06–1.1)				
	**Speaking** **OIDP = 0** **OIDP ≥1**	92 (96.8) 3 (3.2)		173 (91.1) 17 (8.9)		1 1.2 (0.5–1.7)		1 0.9 (0.5–1.9)				
	**Sleeping and relaxing** **OIDP = 0** **OIDP ≥1**	77 (81.1) 18 (18.9)		155 (81.6) 35 (18.4)		1 1.1 (0.6–1.9)		1 0.8 (0.4–1.6)				
	**Smiling and showing teeth** **OIDP = 0** **OIDP ≥1**	20 (21.1) 75 (78.9)		130 (68.4) 60 (31.6)		1 8.4 (4.2–16.5) *p* < 0.001		1 10.9 (4.5–26.6) *p* < 0.001				
	**Maintaining usual emotional state** **OIDP = 0** **OIDP ≥1**	65 (68.4) 30 (31.6)		152 (80.0) 38 (20.0)		1 1.8 (1.1–3.2) *p* < 0.05		1 1.8 (0.9–3.6)				
	**Carrying out school related tasks** **OIDP = 0** **OIDP ≥1**	91 (95.8) 4 (4.2)		168 (88.4) 22 (11.6)		1 0.4 (0.1–1.1)		1 0.5 (0.1–1.8)				
	**Enjoying contact with people** **OIDP = 0** **OIDP ≥1**	32 (33.7) 63 (66.3)		146 (76.8) 44 (23.2)		1 5.6 (3.2–9.8) *p* < 0.001		1 6.1 (3.1–12.1) *p* < 0.001				
	**Overall OIDP** **OIDP = 0** **OIDP ≥1**	11 (11.6) 84 (88.4)		78 (41.1) 112 (58.9)		1 5 (2.4–10.2) *p* < 0.001		1 3.9 (1.6–9.1) *p* < 0.05				
Cortes et al. (2002) [35]	**Variable**	**Cases n (%)**		**Controls n (%)**		**OR-unadjusted** **(CI = 95%)**		**OR-adjusted** **(CI = 95%)**		Effect of individual item on OHRQoL: Eating and enjoying food: Cases were 13.4 times (95% CI = 3.0–61.0) more likely to report an impact for ‘eating and enjoying food’ than children with no traumatic dental injury. Speaking and pronouncing clearly: This item had the least impact for both case and control groups. Cleaning your mouth: The impact for ‘cleaning teeth’ was statistically and significantly associated with the group of children with untreated fractured teeth. Children with fractured teeth were more likely to report an impact for this item than children without TDI. Smiling, laughing, and showing teeth without embarrassment: The most prevalent OIDP impact was seen in this item for both cases and controls. Children with fractured teeth were more likely to report an impact for this item than children without a TDI. The appearance of untreated fractured teeth was the main factor affecting this OIDP item. Maintaining emotional state without being irritable: The impact of this item on OIDP was statistically significant. Cases were 11.8 times more likely to report an impact for this item than controls. The appearance of untreated fractured teeth was the main factor affecting this OIDP item. Contact with people: There was a statistically significant association between ‘enjoying contact with people’ and the presence of fractured teeth. Cases were 10.0 times more likely to report an impact for the item ‘enjoying contact with people’ when compared to controls. Overall OIDP score: Children with fractured teeth were 20 times more likely to report any impact on their daily living than children with no traumatic dental injury. This shows that children with fractured teeth had significantly higher OIDP scores than those without TDI.		
	**Eating and enjoying your food** **OIDP = 0** **OIDP >0**	55 (80.9) 13 (19.1)		134 (98.5) 2 (1.5)		1 13.0 (2.9–57.6) *p* < 0.01		1 13.4 (3.0–61.0) *p* < 0.01				
	**Speaking and pronouncing clearly** **OIDP = 0** **OIDP >0**	64 (94.1) 4(5.9)		135 (99.3) 1 (0.7)		1 8.0 (0.9–71.6)		1 8.0 (0.9–75.0)				
	**Cleaning your mouth** **OIDP = 0** **OIDP >0**	58 (85.3) 10 (7.14)		129 (94.9) 7 (5.1)		1 4.0 (1.2–13.1) *p* < 0.05		1 3.9 (1.2–13.0) *p* < 0.05				
	**Smiling, laughing, and showing our teeth without embarrassment** **OIDP = 0** **OIDP >0**	30 (44.1) 38 (55.9)		118 (86.8) 18 (13.2)		1 8.6 (3.8–19.5) *p* < 0.001		1 15.7 (5.0–44.6) *p* < 0.001				
	**Maintaining your emotional state without being irritatble** **OIDP = 0** **OIDP >0**	45 (66.2) 23 (33.8)		129 (94.9) 7 (5.1)		1 10.4 (3.6–30.3) *p* < 0.001		1 11.8 (3.9–35.5) *p* < 0.001				
	**Contact with people** **OIDP = 0** **OIDP >0**	58 (85.3) 10 (14.7)		134 (98.5) 2 (1.5)		1 10.0 (2.2–45.6) *p* < 0.01		1 10.0 (2.1–47.2) *p* < 0.001				
	**Overall OIDP** **OIDP = 0** **OIDP >0**	23 (33.8) 45 (66.2)		116 (85.3) 20 (14.7)		1 10.2 (4.6–22.8) *p* < 0.001		1 20.0 (7.0–57.7) *p* < 0.001				
Basavaraj et al. (2014) [28]	**OIDP**			**Impact score** **mean (SD)**		**OR (95% CI)**				Effect of gender on oral impact: There was no association between oral impacts and gender. Effect of age on OIDP: There was a significant difference among the age groups, with younger age groups reporting more trauma. Effect of individual item on OIDP: Eating: Children with TDI demonstrated significant impacts on eating (OR = 11.0). Difficulty with eating due to oral problems was the most common impact (72.9%) and led to more severe oral impacts on children’s quality of life than impacts on other performances. Children with TDI demonstrated significant impacts on cleaning teeth (OR = 3.5), emotional status (OR = 10.0), smiling (OR = 15.2), and contact (OR = 13.1). Children with TDI were less likely to have an impact on relaxing (OR = 0.6). Overall impact: Significant association was observed for oral impacts on daily performance with TDI at *p*-value ≤ 0.05. Overall, 60% of children reported at least one impact on quality of life attributed to oral health in the last three months.		
	**Overall impacts(%)**			2.49 (3.92)								
	**Eating**			1.60 (4.49)		11.0 (10.7–9.11) *p* ≤ 0.05						
	**Speaking**			0.10 (0.71)		7.0 (2.1–55.2)						
	**Cleaning teeth**			0.87 (3.07)		3.5 (2.4–16.2) *p* ≤ 0.05						
	**Relaxing**			0.39 (2.65)		0.6 (0.1–0.9)						
	**Emotion**			0.61 (2.92)		10.0 (1.2–18.2) *p* ≤ 0.05						
	**Smiling**			1.31 (4.22)		15.2 (11.1–24.2) *p* ≤ 0.05						
	**Study**			0.06 (0.59)		0.8 (0.2–1.7)						
	**Contact**			0.51 (2.13)		13.1 (9.4–19.2) *p* ≤ 0.05						
Ramos-Jorge et al. (2007) [36]	**Variables**			**Case group (n%)**		**Control group (n%)**				** *p* **	Effect of mother’s education on OHRQoL: There were no statistical significant differences between the case and control group with relation to mother’s education. Adolescents with aesthetically-treated enamel dentin fractures had a greater chance of presenting an oral impact on daily activities than those never having suffered dental trauma (OR 3.26–95% (CI) = 1.4–7.7). The treatment of coronary fractures did not eliminate the impact of trauma on the daily life of the adolescents surveyed, although it possibly reduced such an impact.	
	**Eating** **OIDP = 0** **OIDP ≥1**			34 (85.0) 6 (15.0)		153 (95.6) 7 (4.4)				0.015 C C- chi-square test		
	**Speaking** **OIDP = 0** **OIDP ≥1**			38 (95.0) 2 (5.0)		158 (98.8) 2 (1.2)				0.179 F F- Fisher’s exact test		
	**Showing teeth** **OIDP = 0** **OIDP ≥1**			26 (65.0) 14 (35.0)		138 (86.3) 22 (13.8)				0.002 C C- chi-square test		
	**Maintaining stable emotional state** **OIDP = 0** **OIDP ≥1**			39 (97.5) 1 (2.5)		160 (100.0) 0 (0.0)				0.215 F F- Fisher’s exact test		
	**Cleaning mouth** **OIDP = 0** **OIDP ≥1**			40 (100.0) 0 (0.0)		159 (99.5) 1 (0.5)				0.215 F		
	**Total OIDP** **OIDP = 0** **OIDP ≥1**			24 (60.0) 16 (40.0)		133 (83.1) 27 (16.9)				0.001 C		

*, **, a, b, † and ‡: Represents significant values obtained within the data set.

## Data Availability

Not applicable.

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
