# Peer review of "Oral Health-Related Quality of Life in Children and Adolescents with a Traumatic Injury of Permanent Teeth and the Impact on Their Families: A Systematic Review"

_ijerph, 2022, doi:10.3390/ijerph19053087_

Round 1

Reviewer 1 Report

Dear Editor,

thank you for considering me as a reviewer for this paper.

I found the topic interesting and the research well designed, nevertheless I have some suggestions for Authors:

1- generally speaking the results of the research are not adequately presented in the related manuscript section. Several papers included in the review considered the same variable as potentially affecting OHRQoL as sex, being injuries treated or not, socioeconomic status, education level/school, etc.  Authors summarized in few lines these reports in a table but did not offer a table which indicates how many previous articles found correlation with this or that variable. This further table will be largely useful to reader and will support better the discussion of results . The figure no.6 is less informative and it can be erased if a descriptive table as suggested above should be provided.

2- I'm not sure that it makes sense to perform a mata analysis based on 2 or 3 papers (papers 34-36) as Authors did.

3- Some conclusions should be better explained. Authors stated : "Since the majority of studies used well validated questionnaire tools and were of high quality, it can be concluded that TDI impact on OHRQoL is significant". It seems that the impact of QHRQoL was already assessed by previous studies,  the  previous scientific evidence relies upon the use of validated questionnaires and then the present review does not offer any novelty  nor for correlation between TDI and QHRQoL neither for methods used to assess the quality of the research tools.

4- in this view (see point # 3) the aims of the paper could be reconsidered.  The actual aims are : "Therefore objective of this systematic review is to assess the impact of traumatic dental injury of permanent teeth on oral health related quality of life". Maybe at least a second aim can be included looking at the advantages of the experimented : "Use of tools like the Q-SSP checklist to evaluate study quality, will raise the profile of reporting standards and drive greater precision in the reporting of survey study methods".

5- Discussion. This section will benefit from a more extensive report about results as suggested above (comment #1). For instance discussion actually provideds several sentence  as : " Most common tool used for the assessment of OHRQoL was CPQ(11-14) in 10 studies [15,22,28–31,33,41–43]" or " However, although 4 studies [21,28,29,31] 274
out of 8 [21,26–31,41] showed no association between gender and its impact on OHRQoL". These sentences require the reader to reconsider the references/original articles to verify the discussed issue. In this sense che review presented here tends to be useless. I strogly suggest to summarize results as explained above, hence the discussion can be based upon the findings of the review results.

6- the paper should be submitted for review to a professional English proof reading services.

Author Response

Reviewer 1

Response:

Dear Reviewer,

We are very thankful for all your valuable comments. All your suggestions were addressed accordingly: the line is given (in Track Changes - Final: Show Markup) and the changed text.

Dear Editor,

thank you for considering me as a reviewer for this paper.

I found the topic interesting and the research well designed, nevertheless I have some suggestions for Authors:

1- generally speaking the results of the research are not adequately presented in the related manuscript section. Several papers included in the review considered the same variable as potentially affecting OHRQoL as sex, being injuries treated or not, socioeconomic status, education level/school, etc.  Authors summarized in few lines these reports in a table but did not offer a table which indicates how many previous articles found correlation with this or that variable. This further table will be largely useful to reader and will support better the discussion of results. The figure no.6 is less informative and it can be erased if a descriptive table as suggested above should be provided.

Response: Thank you for your comment. We have added Table 4, which shows correlations between different variables considered by studies included in the review. Also, as suggested by the Reviewer, Figure 6 was removed.

2- I'm not sure that it makes sense to perform a meta-analysis based on 2 or 3 papers (papers 34-36) as Authors did.

Response: Thank you for your comment. We agree and the meta-analysis was removed.

3- Some conclusions should be better explained. Authors stated: "Since the majority of studies used well validated questionnaire tools and were of high quality, it can be concluded that TDI impact on OHRQoL is significant". It seems that the impact of QHRQoL was already assessed by previous studies, the previous scientific evidence relies upon the use of validated questionnaires and then the present review does not offer any novelty nor for correlation between TDI and QHRQoL neither for methods used to assess the quality of the research tools.

Response: Thank you for your comment. The present investigation is based on OHRQoL based on trauma to permanent teeth specifically. Previous reviews were either on deciduous dentition or mixed dentition. Secondly, we have used new tool to assess the studies. These two factors, in our opinion, make the study unique.

4- in this view (see point # 3) the aims of the paper could be reconsidered.  The actual aims are: "Therefore objective of this systematic review is to assess the impact of traumatic dental injury of permanent teeth on oral health related quality of life". Maybe at least a second aim can be included looking at the advantages of the experimented: "Use of tools like the Q-SSP checklist to evaluate study quality, will raise the profile of reporting standards and drive greater precision in the reporting of survey study methods".

Response: Thank you for your comment. The manuscript was corrected following your suggestion and the following was added in introduction: (lines 78-80):

“Therefore objective of this systematic review is to assess the impact of traumatic dental injury of permanent teeth on oral health related quality of life and assess the study quality using Q-SSP checklist.”

Also detailed information on the quality assessment of included studies was added (line 129-137):

“The quality was judged for each domain and is expressed as percentage by dividing YES (Y) scores by the Total (T) number of APPLICABLE items and multiplying by 100. When (T) = 20, then a score of Y/T ≥ 75% may be considered acceptable quality. When (T) = 19, then a score of Y/T ≥ 73% may be considered acceptable quality. When (T) = 18, then a score of Y/T ≥ 72% may be considered acceptable quality. When (T) = 17, then a score of Y/T ≥ 70% may be considered acceptable quality. If report fails to attain a Y score for 5 of the items, then it may be classed as of questionable quality. The assessment was added to an Excel spreadsheet and then imported into ROBVIS (Risk of Bias Visualization Software).”

5- Discussion. This section will benefit from a more extensive report about results as suggested above (comment #1). For instance discussion actually provided several sentence  as: " Most common tool used for the assessment of OHRQoL was CPQ(11-14) in 10 studies [15,22,28–31,33,41–43]" or "However, although 4 studies [21,28,29,31] out of 8 [21,26–31,41] showed no association between gender and its impact on OHRQoL". These sentences require the reader to reconsider the references/original articles to verify the discussed issue. In this sense the review presented here tends to be useless. I strongly suggest to summarize results as explained above, hence the discussion can be based upon the findings of the review results.

Response: Thank you for your comment. We have summarized the results in Table 4.

6- the paper should be submitted for review to a professional English proof reading services.

Response: Thank you for your comment. Professional English proof reading was done.

Reviewer 2 Report

Review report on ‘Oral health related quality of life in children and adolescent with traumatic injury of permanent teeth and impact on their families. A systematic review and meta- analysis.’

Dear Authors,

Congratulations on the hard work and dedicationyou put in this project. The manuscript is extremely well-written and structured. A comprehensive analysis of the parameters taken into consideration is performed and all the keypoints are explored in-depth.

Although, in order to make it suitable for publication, I believe a further expansion and detailing of the conclusion section needs to be made, so that it can reflect the analysis and the results of impact obtained. 

Author Response

Reviewer 2

Review report on ‘Oral health related quality of life in children and adolescent with traumatic injury of permanent teeth and impact on their families. A systematic review and meta- analysis.’

Dear Authors,

Congratulations on the hard work and dedication you put in this project. The manuscript is extremely well-written and structured. A comprehensive analysis of the parameters taken into consideration is performed and all the key points are explored in-depth.

Although, in order to make it suitable for publication, I believe a further expansion and detailing of the conclusion section needs to be made, so that it can reflect the analysis and the results of impact obtained. 

Response:

Dear Reviewer,

We are very thankful for your kind words and all your comments.

We have added an extensive Table 4 with inference of results.

The conclusion section was revised as follows (lines 355-364):

“Traumatic injuries to permanent dentition affects both a child and their caregivers or parents. These injuries affects both the genders, however adolescent girls tend to have more negative impact on their OHRQOL compared boys. TDI and it’s severity significantly affects the social and emotional well-being of children and their families. Parents education and socioeconomic status plays a major role in providing care and treatment of TDI in children. Treatments of TDI not only improves the esthetic and functional aspects of dentition but also enhances the OHRQOL. Since the majority of studies used well validated questionnaire tools and were of high quality, it can be concluded that TDI impact on OHRQoL is significant.”

Reviewer 3 Report

Thank you for the opportunity to review this paper. You have taken on the difficult task of a meta-analysis of OHRQOL in children.

Due to the difficulty of the topic, it is understandable that there are few papers on the target.
However, with only two studies, it is not a meta-analysis. In particular, the funnel plot for heterogeneity cannot be judged.

At the very least, please provide the results of Eggaer's regression method or Begg's Kendall rank correlation coefficient instead of the visual judgment of the funnel plot.

Please consider it.

Author Response

Reviewer 3

Thank you for the opportunity to review this paper. You have taken on the difficult task of a meta-analysis of OHRQOL in children.

Due to the difficulty of the topic, it is understandable that there are few papers on the target.
However, with only two studies, it is not a meta-analysis. In particular, the funnel plot for heterogeneity cannot be judged.

At the very least, please provide the results of Eggaer's regression method or Begg's Kendall rank correlation coefficient instead of the visual judgment of the funnel plot.

Response:

Dear Reviewer,

We are very thankful for your kind words and all your valuable comments. We fully agree with your opinion. Following your suggestion the meta-analysis was removed and Figure 6 was erased. Instead, extensive Table 4 with inference of results of included studies was added.

Reviewer 4 Report

Deviations/changes from the registered PROSPERO protocol should be explained in the manuscript. For example, the protocol specified that Google Scholar was to be searched; however, this is not mentioned in the manuscript.

The abstract states that the “Cochrane Manual of Systematic Reviews of Interventions” was used to assess the risk of bias. However, this is not mentioned or cited in the Methods section.

Please clarify why an assessment for publication bias was not performed.

In the Methods section, please clarify how weighting was performed for the meta-analysis. Was a random-effects or fixed-effects model used?

AMSTAR assessments evaluate whether systematic reviews have provided a list of articles excluded at the full document screening stage, as well as the reason for exclusion of each study. I would suggest that you include this.

Some of the meta-analysis results were associated with a high degree of heterogeneity. Please comment on this in the Discussion section.

Systematic review quality assessments often evaluate whether authors have reviewed the potential impacts of funding for each included study. I would suggest that you review this element.

Author Response

Reviewer 4

Dear Reviewer,

We are very thankful for all your comments. All your suggestions were addressed accordingly: the line is given (in Track Changes - Final: Show Markup) and the changed text.

Deviations/changes from the registered PROSPERO protocol should be explained in the manuscript. For example, the protocol specified that Google Scholar was to be searched; however, this is not mentioned in the manuscript.

Response: Thank you for your comment. We mentioned it in revised manuscript, line 93.

“A comprehensive electronic search for relevant articles was performed in the following databases: PubMed, Cochrane Library, MEDLINE, Google Scholar.”

The abstract states that the “Cochrane Manual of Systematic Reviews of Interventions” was used to assess the risk of bias. However, this is not mentioned or cited in the Methods section. Please clarify why an assessment for publication bias was not performed. In the Methods section, please clarify how weighting was performed for the meta-analysis. Was a random-effects or fixed-effects model used?

Response: Thank you for your comment. We removed meta-analysis from the study as only 2 articles had similar parameters. Therefore publication bias was also not included in the revised manuscript. The correction has been made in the Abstract section and the statement about risk of bias removed.

AMSTAR assessments evaluate whether systematic reviews have provided a list of articles excluded at the full document screening stage, as well as the reason for exclusion of each study. I would suggest that you include this.

Response: Thank you for your comment. The justification of exclusion of articles at the full document screening is given in Figure 2. Also, following your suggestion, Table A1 containing the list of excluded studies have been provided in Appendix section.

Some of the meta-analysis results were associated with a high degree of heterogeneity. Please comment on this in the Discussion section.

Response: Thank you for your comment. Following Reviewers’ suggestion, the meta-analysis was removed from the study, so we made a note of the high degree of heterogeneity of the included studies in the manuscript. Under section synthesis of result we have mentioned as follows (lines 211-212):

“Due to a high heterogeneity of the data, it was not possible to perform meta-analysis for all the parameters used in the included studies therefore qualitative assessment was done.”

Systematic review quality assessments often evaluate whether authors have reviewed the potential impacts of funding for each included study. I would suggest that you review this element.

Response: Thank you for your comment. The information about the funding of each included study was added in Table 3.

Round 2

Reviewer 1 Report

Authors made all requested changes. Table 4 is too long compared to the overall manuscript, maybe it can be summarized during editing process.

Author Response

Response: We would like to thank the Reviewer for their time and effort in reviewing our article, and for all valuable comments that helped to improve our manuscript. We have edited Table 4. We strongly believe, that the Journal's Editorial Team will help us with editing it further.

Reviewer 3 Report

It must have been a difficult task to make so many revisions in such a short period of time. The authors responded sincerely to my request. The quality of this paper has been greatly improved by withdrawing the meta-analysis and making it a review manuscript.

Author Response

Response: We would like to thank the Reviewer for their time and effort in reviewing our article, and for all valuable comments that helped to improve our manuscript.

Reviewer 4 Report

Thank you for addressing the previous comments. The manuscript is now much improved.

Author Response

(The authors gave the same response as above.)
